# CTRL&SHIFT: High-quality Geometry-Aware Object Manipulation in Visual Generation

**Penghui Ruan**[1,2,*]    **Bojia Zi**[3,*]    **Xianbiao Qi**[4,†]    **Youze Huang**[5]    **Rong Xiao**[4]

**Pichao Wang**[6,‡]    **Jiannong Cao**[1,§]    **Yuhui Shi**[2,§]

[1]The Hong Kong Polytechnic University    [2]Southern University of Science and Technology

[3]The Chinese University of Hong Kong    [4]IntelliFusion Inc.

[5]University of Electronic Science and Technology of China    [6]NVIDIA

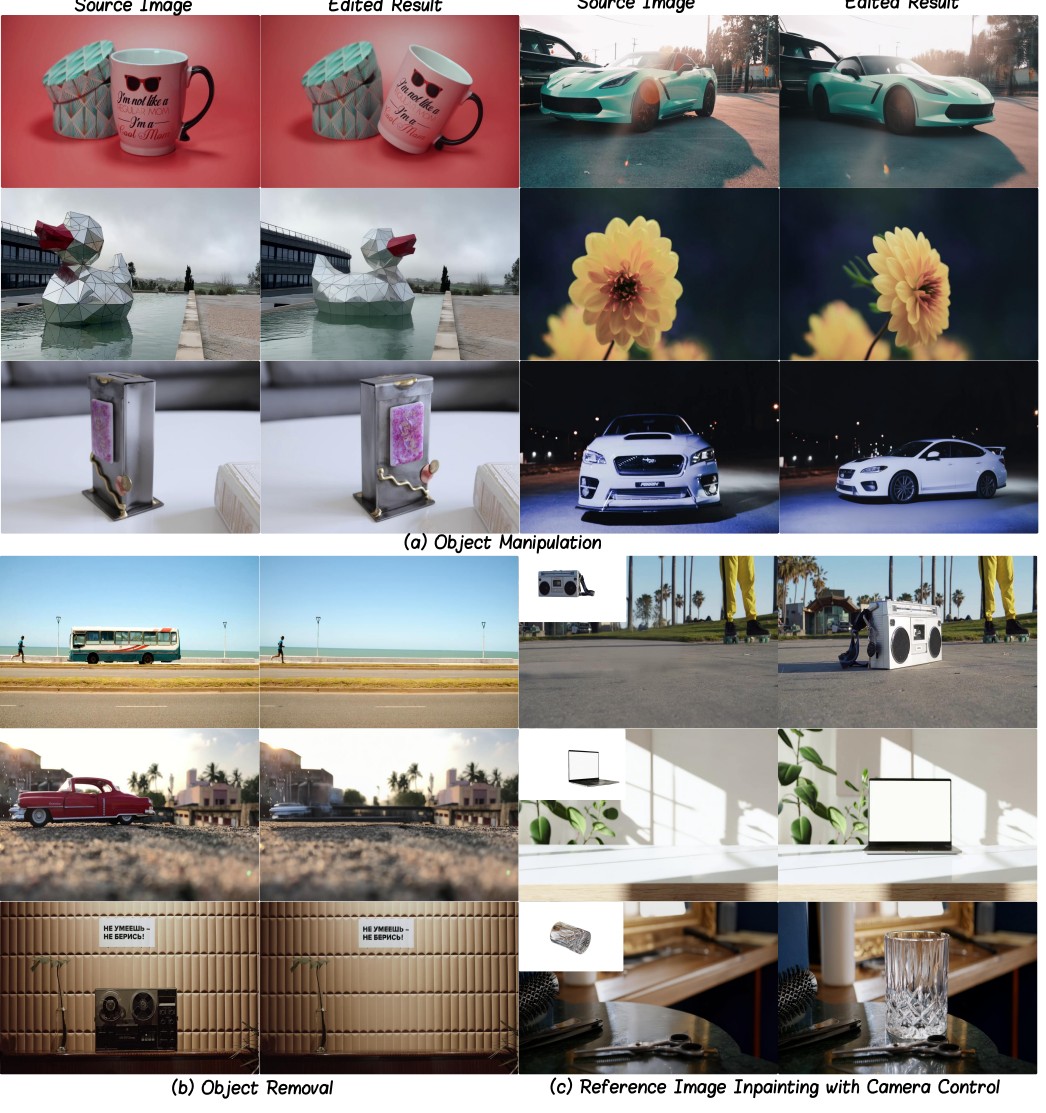

Figure 1: **Results of Ctrl&Shift. Ctrl&Shift** demonstrates its superior capability (controllability, plausibility and consistency) on tasks including (a) precise object manipulation, (b) visual object removal, and (c) reference image inpainting with precise camera pose control.

[*] Equal Contribution    [†] Project Lead    [‡] This work is not related to the author's position at NVIDIA.
[§] Corresponding Authors

ABSTRACT

Object-level manipulation—relocating or reorienting objects in images or videos while preserving scene realism—is central to film post-production, AR, and creative editing. Yet existing methods struggle to jointly achieve three core goals: background preservation, geometric consistency under viewpoint shifts, and user-controllable transformations. Geometry-based approaches offer precise control but require explicit 3D reconstruction and generalize poorly; diffusion-based methods generalize better but lack fine-grained geometric control. We present **Ctrl&Shift**, an end-to-end diffusion framework to achieve geometry-consistent object manipulation without explicit 3D representations. Our key insight is to decompose manipulation into two stages—object removal and reference-guided inpainting under explicit camera pose control—and encode both within a unified diffusion process. To enable precise, disentangled control, we design a multi-task, multi-stage training strategy that separates background, identity, and pose signals across tasks. To improve generalization, we introduce a scalable real-world dataset construction pipeline that generates paired image and video samples with estimated relative camera poses. Extensive experiments demonstrate that **Ctrl&Shift** achieves state-of-the-art results in fidelity, viewpoint consistency, and controllability. *To our knowledge, this is the first framework to unify fine-grained geometric control and real-world generalization for object manipulation—without relying on any explicit 3D modeling.*

## 1 INTRODUCTION

Object-level manipulation—such as relocating or rotating an object within a single image or video while preserving the surrounding scene—is a fundamental primitive in film post-production, augmented reality (AR), and creative visual editing. For instance, adjusting the placement of a prop in a movie frame or changing the camera angle of a product in an AR preview both demand geometry-aware edits that remain photorealistic under viewpoint changes. Failing to preserve geometric consistency leads to warped objects, unnatural shadows, or background artifacts—breaking immersion and rendering the edit unusable in professional workflows.

Despite recent progress in visual editing, controllable and consistent object manipulation remains a persistent challenge. At the heart of this difficulty lies a fundamental trade-off: **geometry-based methods offer precise control but poor generalization**, while **diffusion-based methods generalize well but lack fine-grained control over geometry**. Geometry-based approaches (Mildenhall et al., 2020; Kerbl et al., 2023; Michel et al., 2023; Chen et al., 2025; Yenphraphai et al., 2024) rely on multi-view optimization or explicit 3D reconstruction to maintain consistency across views. However, these techniques often require synthetic data or per-scene optimization, limiting scalability and realism. On the other hand, diffusion-based editors (Zhang & Agrawala, 2023; Wu et al., 2024; Jiang et al., 2025) have demonstrated remarkable generalization to in-the-wild content, enabling free-form edits via text prompts or trajectories (Choi et al., 2023). Yet, these models struggle to deliver precise control over object pose, making it difficult to specify fine-grained geometric transformations.

This paper introduces **Ctrl&Shift**, a new framework that **breaks this trade-off**: it enables geometry-consistent, fine-grained object manipulation with strong generalization to real-world content—*without requiring explicit 3D reconstruction at inference*. This marks a conceptual shift: rather than lifting content into 3D space to perform edits, we inject precise viewpoint control directly into a 2D diffusion process. Our core insight is to decompose object manipulation into two sub-tasks—*object removal* and *reference-guided inpainting under explicit camera pose*—which can be jointly modeled within a unified diffusion framework. **Disentangled Control via Multi-Task Training.** The central challenge is enabling the model to respond coherently to multiple conditioning signals: background context, object identity, spatial masks, and relative viewpoint changes. To address this, we design a unified conditioning interface and adopt a multi-task training scheme aligned with the natural task decomposition: one task teaches object removal, another reference-conditioned inpainting with camera control, and a third combines both for full manipulation. This structure ensures each signal plays a clear, interpretable role during training and inference. **From**

**Synthetic Pretraining to Real-World Generalization.** While multi-task learning helps disentangle functional roles, achieving realistic geometry-aware control in natural imagery requires high-quality, pose-annotated supervision. Synthetic datasets often fall short in realism and diversity, and existing real-world data rarely includes accurate camera annotations. To close this gap, we develop a scalable pipeline for constructing paired image and video samples from real content, enriched with estimated relative camera poses. Our method leverages 3D object reconstruction, differentiable camera pose estimation, and harmonized rendering to synthesize target views under novel camera configurations—enabling large-scale training on realistic data with geometric supervision.

**Contributions.** This work introduces a new approach to controllable object manipulation that fuses the structural precision of geometry-based pipelines with the scalability of generative diffusion, all within a unified 2D framework. Our contributions are:

- **Conceptual Innovation.** We propose **Ctrl&Shift**, an end-to-end framework to achieve geometry-consistent object manipulation without requiring explicit 3D representations at inference, by injecting relative camera pose control directly into the diffusion process.
- **Architectural Design.** We disentangle background, object identity, spatial masks, and geometric transformations through a multi-task, multi-stage training strategy that mirrors the semantic structure of object manipulation.
- **Data Construction Pipeline.** We introduce a scalable method to create real-world paired supervision with accurate camera control, by reconstructing object meshes, estimating poses via differentiable rendering, and harmonizing novel views through learned object pasting model.
- **Systematic Benchmarking.** We evaluate **Ctrl&Shift** across multiple datasets and introduce GeoEditBench, a new benchmark for geometry-aware editing. Our model consistently achieves state-of-the-art results in fidelity, controllability, and viewpoint consistency. Moreover, by avoiding 3D modeling at inference while retaining precise geometric control, our approach bridges geometry-based rigor with diffusion flexibility, enabling scalable, controllable editing in the wild.

## 1.1 RELATED WORK

**Diffusion-based Object Editing.** Diffusion models have enabled powerful visual editing capabilities (Rombach et al., 2022; Saharia et al., 2022; Wu et al., 2025; Zi et al., 2025b). *Text-conditioned approaches offer flexibility but lack precise control over spatial transformations, often leading to ambiguous object manipulation.* Trajectory-based methods convert motion signals into spatial conditioning via architectures like ControlNet (Zhang & Agrawala, 2023; Jiang et al., 2025; Wu et al., 2024). For instance, DragAnything (Wu et al., 2024) enables interactive point-based manipulation, while FateZero (Qi et al., 2023) and Tune-A-Video (Wu et al., 2023) extend editing to the temporal domain. Object insertion methods like ObjectAdd (Zhang et al., 2024) and InVi (Saini et al., 2024) modify diffusion processes for adding objects without retraining. However, without explicit 3D representations, these methods struggle to provide geometric control, particularly for viewpoint changes, resulting in object identity drift, background corruption, and inconsistent camera.

**Geometry-aware Object Editing.** Complementary approaches leverage 3D supervision to maintain multi-view consistency. Single-image 3D lifting methods like Zero-1-to-3 (Liu et al., 2023) and SyncDreamer (Liu et al., 2024) synthesize novel views but require harmonization for real scenes. Dataset-driven methods using synthetic data face domain gap challenges (Reizenstein et al., 2021; Deitke et al., 2022). Recent works integrate 3D information into diffusion: GeoDiffuser uses geometry-based conditioning for precise edits (Sajnani et al., 2025), Diffusion Handles enables handle-based 3D-aware manipulation (Pandey et al., 2023), and OBJect 3DIT provides language-guided multi-view consistent editing Michel et al. (2023). ObjectMover leverages video priors for object movement, while BlenderFusion uses 3D engines for manipulation (Yu et al., 2025; Chen et al., 2025). However, these methods typically require explicit 3D reconstruction and manipulation.

## 2 METHOD

The primary goal of object manipulation is to shift and rotate the object according to the relative pose signals while maintaining visual coherence. In our framework, we formulate this as an image/video editing task—specifically, given: the source video frames $\mathbf{X}^{\text{src}} \in \mathbb{R}^{T \times 3 \times H \times W}$, which provide the

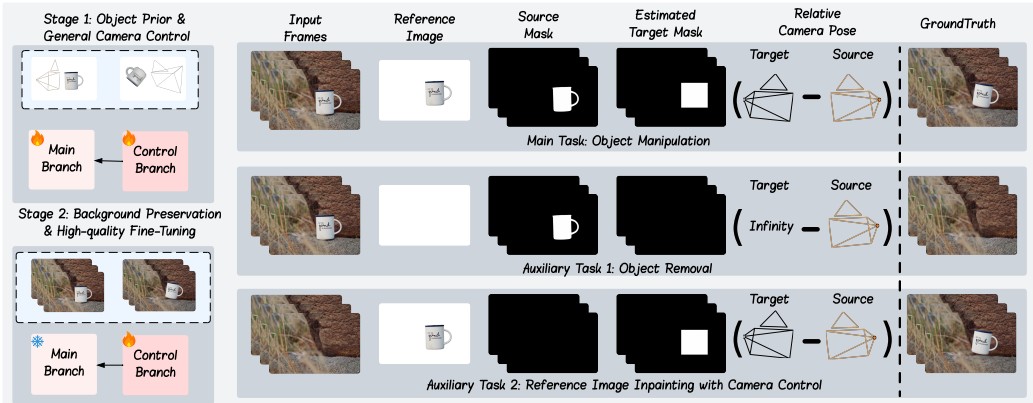

Figure 2: The proposed **Ctrl&Shift** framework employs a multi-task, multi-stage training paradigm integrating object manipulation, removal, and reference inpainting with explicit camera control. Stage 1 focuses on acquiring object priors and camera control; Stage 2 emphasizes background preservation through fine-tuning on high-quality data.

background context; the reference object image $\mathbf{I}^{\text{ref}} \in \mathbb{R}^{3 \times H \times W}$ (extracted from the first source frame), which disambiguates the object identity; the source mask $\mathbf{M}^{\text{src}} \in \{0, 1\}^{T \times 1 \times H \times W}$, which delineates the region to be removed; the estimated target mask $\hat{\mathbf{M}}^{\text{tgt}} \in \{0, 1\}^{T \times 1 \times H \times W}$ (as detailed in Section 2.1), which specifies the coarse region for object placement; the relative camera-pose descriptor $\mathbf{f} \in \mathbb{R}^8$ (as defined in Section 2.2), which encodes the geometric transformation between the source and target viewpoints. Our model translates and rotates the object to the target location/viewpoint and predicts target frames $\mathbf{X}^{\text{tgt}} \in \mathbb{R}^{T \times 3 \times H \times W}$.

**Ctrl&Shift** adopts a ControlNet-style DiT architecture with hidden size $d_{\text{model}} = 1536$. The inputs are processed into a unified latent space: source frames $\mathbf{X}^{\text{src}}$ and reference $\mathbf{I}^{\text{ref}}$ are encoded via $\mathcal{E}_{\text{VAE}}$, while masks $\mathbf{M}^{\text{src}}$ and $\hat{\mathbf{M}}^{\text{tgt}}$ are downsampled via pixel unshuffle $\Pi_s$. In the control branch, these features are concatenated channel-wise, temporally replicating $\mathbf{I}^{\text{ref}}$ to match the video frames, then projected and injected into the DiT blocks via zero-initialized convolutions. The main backbone operates on noised latents derived from the concatenated ground-truth video $\mathbf{X}^{\text{tgt}}$ and reference $\mathbf{I}^{\text{ref}}$. To enforce geometric control, the relative-pose descriptor $\mathbf{f}$ is mapped by a 3-layer MLP $\mathcal{E}_{\text{cam}}$ to embeddings that are injected via cross-attention, guiding the viewpoint while preserving identity and background context.

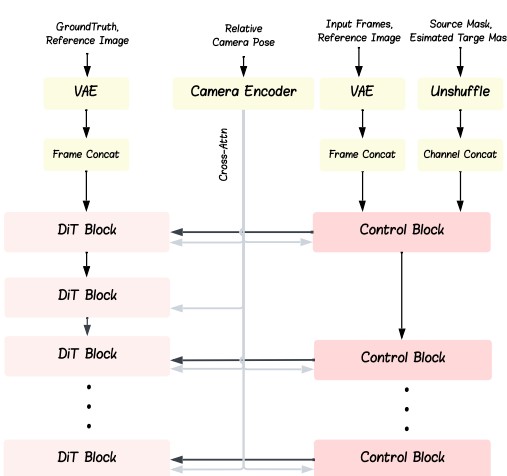

Figure 3: Overview of the architecture.

## 2.1 MASK ENCODING

We cast the object manipulation as object removal and reference image inpainting tasks. During training we provide a source mask $\mathbf{M}^{\text{src}}$ and an *estimated* target mask $\hat{\mathbf{M}}^{\text{tgt}}$. As the true target mask is unknown during inference, we obtain $\hat{\mathbf{M}}^{\text{tgt}}$ from $\mathbf{M}^{\text{src}}$ by: (i) squaring/tightening $\mathbf{M}^{\text{src}}$ via its bounding box; (ii) scaling by the distance ratio $\frac{d^{\text{src}}}{d^{\text{tgt}}}$ to approximate apparent size, (iii) shifting on the image plane by $(\Delta r_x, \Delta r_y)$; and (iv) truncating outside the frame.

**Mask Semantics and Representation.** In our setting, a value of $1$ indicates a region to be *edited/painted* and $0$ indicates a region to be *preserved*; these semantics differ from image intensities (e.g., white vs. black color). To preserve this binary meaning and avoid treating masks as

appearance, we do *not* encode masks with the VAE. Instead, we map masks to the VAE latent grid using a space-to-depth (pixel–unshuffle) operation that exactly matches the VAE stride. Concretely, for spatial stride $s$, and temporal stride $t$, we transform

$$\mathbf{M} \in \{0,1\}^{T \times 1 \times H \times W} \quad \longmapsto \quad \Pi_s(\mathbf{M}) \in \{0,1\}^{\frac{T}{t} \times ts^2 \times \frac{H}{s} \times \frac{W}{s}},$$

which reduces resolution while increasing channels. This procedure preserves the mask's binary structure, avoids semantic confusion with RGB values, and provides high-fidelity, stride-aligned guidance to the model.

## 2.2 CAMERA POSE ENCODING

In this subsection, we will introduce how we obtain the relative pose descriptor $\mathbf{f}$. We position the object at the world origin and use a *look-at* camera oriented toward the origin with a fixed world-up vector $\mathbf{u}_w = (0,1,0)^\top$. Each view is parameterized by

$$\mathbf{s} = \big(\text{yaw}, \text{ pitch}, d, r_x, r_y\big)^\top,$$

where yaw $\in (-\pi, \pi]$ is the azimuth about the $+Y$-axis (zero along $+X$, increasing toward $+Z$), pitch $\in [-\pi/2, \pi/2]$ is the elevation above the $XZ$-plane, $d > 0$ is the camera-origin distance, and $(r_x, r_y) \in [-1,1]^2$ are normalized-device-coordinate (NDC) shifts applied post-projection, with $(0,0)$ at the image center. For source and target views $\mathbf{s}^{\text{src}}$ and $\mathbf{s}^{\text{tgt}}$, we compute world-to-camera extrinsics via

$$(\mathbf{R}_{\text{src}}, \mathbf{t}_{\text{src}}) = \text{LOOKAT}\big(\text{yaw}^{\text{src}}, \text{pitch}^{\text{src}}, d^{\text{src}}; \mathbf{u}_w\big),$$

$$(\mathbf{R}_{\text{tgt}}, \mathbf{t}_{\text{tgt}}) = \text{LOOKAT}\big(\text{yaw}^{\text{tgt}}, \text{pitch}^{\text{tgt}}, d^{\text{tgt}}; \mathbf{u}_w\big),$$

where LOOKAT positions the camera at $\big(d\cos(\text{pitch})\cos(\text{yaw}), d\sin(\text{pitch}), d\cos(\text{pitch})\sin(\text{yaw})\big)^\top$, oriented toward the origin with up-vector $\mathbf{u}_w$, yielding $(\mathbf{R}, \mathbf{t})$ such that $\mathbf{x}_c = \mathbf{R}\mathbf{x}_w + \mathbf{t}$. The relative transform matrix and the translation vector from source-camera to target-camera coordinates are individually calculated as

$$\mathbf{R}_{\text{rel}} = \mathbf{R}_{\text{tgt}}\mathbf{R}_{\text{src}}^\top, \qquad \mathbf{t}_{\text{rel}} = \mathbf{t}_{\text{tgt}} - \mathbf{R}_{\text{rel}}\mathbf{t}_{\text{src}},$$

where $\mathbf{x}_{\text{tgt}} = \mathbf{R}_{\text{rel}}\mathbf{x}_{\text{src}} + \mathbf{t}_{\text{rel}}$. The relative rotation is encoded via its axis-angle representation

$$\text{aa}(\mathbf{R}_{\text{rel}}) \triangleq \text{vee}\big(\log \mathbf{R}_{\text{rel}}\big) \in \mathbb{R}^3,$$

where $\log \mathbf{R}_{\text{rel}} \in \mathfrak{so}(3)$ is the matrix logarithm and vee : $\mathfrak{so}(3) \to \mathbb{R}^3$ extracts the vector form. The relative NDC shifts are computed as

$$\Delta r_x = r_x^{\text{tgt}} - r_x^{\text{src}}, \qquad \Delta r_y = r_y^{\text{tgt}} - r_y^{\text{src}}.$$

The relative-pose descriptor is

$$\mathbf{f} = \big(\text{aa}(\mathbf{R}_{\text{rel}}); \mathbf{t}_{\text{rel}}; \Delta r_x; \Delta r_y\big)^\top \in \mathbb{R}^8.$$

Each component of $\mathbf{f}$ undergoes a Fourier positional encoding, followed by lightweight MLPs, to create 8 tokens ($d = 4096$). Note that we opt to encode the relative camera pose rather than absolute poses, because defining a canonical absolute pose is challenging across diverse objects. Moreover, during inference, specifying the target pose without a clear baseline is even more difficult, potentially leading to inconsistent and unintuitive controls. In contrast, relative encoding leverages the camera pose from the input frames as an implicit base, requiring only intuitive, drag-like adjustments.

## 2.3 MULTI-TASK MULTI-STAGE TRAINING

**Ctrl&Shift** employs five distinct conditioning signals, integrated through tailored mechanisms: video-like signals are incorporated via ControlNet, while camera control is facilitated through cross-attention mechanisms. Each conditioning signal serves a unique role in the manipulation task. To enhance the model's ability to disambiguate and leverage these signals effectively, we introduce a multi-task training strategy that isolates the functional contributions of each signal. Our tasks are defined as follows:

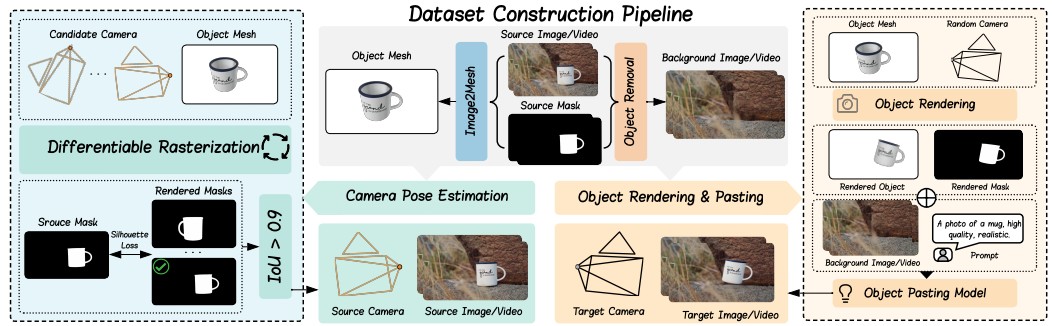

Figure 4: Construction of data pairs $(\mathbf{X}^{\text{src}}, \mathbf{s}^{\text{src}})$ and $(\mathbf{X}^{\text{tgt}}, \mathbf{s}^{\text{tgt}})$. From $\mathbf{X}^{\text{src}}$, an image-to-mesh model reconstructs the object mesh, and $\mathbf{s}^{\text{src}}$ is estimated via differentiable rasterization. The target pose $\mathbf{s}^{\text{tgt}}$ is sampled, the object is rendered using the mesh, and an object pasting model generates $\mathbf{X}^{\text{tgt}}$. Our pipeline supports both image and video data synthesis, as the object pasting model is a reference-image inpainting model capable of editing both image and video. For video inputs, the image-to-mesh reconstruction, camera pose estimation, and rendering are all performed on the first frame.

**Main Task: Object Manipulation.** The primary task of object manipulation involves relocating the foreground object from its source position and viewpoint to a specified target position and viewpoint, while preserving both the object's identity and the background context. Specially, let $\mathcal{E}_{\text{VAE}}$ be a pretrained VAE encoder, and let $\mathcal{E}_{\text{cam}}$ denotes the camera pose encoder. We define the conditioning tuple explicitly via the encoding operators:

$$\mathbf{c}_{\text{main}} = \Big( \mathcal{E}_{\text{VAE}}(\mathbf{X}^{\text{src}}),\ \mathcal{E}_{\text{VAE}}(\mathbf{I}^{\text{ref}}),\ \Pi_s(\mathbf{M}^{\text{src}}),\ \Pi_s(\hat{\mathbf{M}}^{\text{tgt}}),\ \mathcal{E}_{\text{cam}}(\mathbf{f}) \Big), \qquad \mathbf{X}^{\text{tgt}}_{\text{main}} = \mathbf{X}^{\text{tgt}},$$

where $\mathcal{E}_{\text{VAE}}$ denotes the VAE encoder mapping images to latent space, $\Pi_s$ is the space-to-depth operation aligning masks with the VAE latent grid, and $\mathcal{E}_{\text{cam}}$ is the camera pose encoder. The supervision is provided by the ground-truth target frames $\mathbf{X}^{\text{tgt}}$.

**Auxiliary Task 1: Object Removal.** This auxiliary task focuses on removing the foreground object from the source frames while reconstructing a coherent background. To achieve this within our unified conditioning interface, we set the reference image to a pure white image (indicating no object identity), the estimated target mask to all zeros (indicating no region to inpaint), and the relative camera pose $\tilde{\mathbf{f}}$ such that the object is positioned outside the image frame, resulting in $\hat{\mathbf{M}}^{\text{tgt}} = \mathbf{0}$.[*] The conditioning vector is

$$\mathbf{c}_{\text{aux}_1} = \Big( \mathcal{E}_{\text{VAE}}(\mathbf{X}^{\text{src}}),\ \mathcal{E}_{\text{VAE}}(\mathbf{1}),\ \Pi_s(\mathbf{M}^{\text{src}}),\ \Pi_s(\mathbf{0}),\ \mathcal{E}_{\text{cam}}(\tilde{\mathbf{f}}) \Big), \qquad \mathbf{X}^{\text{tgt}}_{\text{aux}_1} = \mathbf{X}^{\text{bg}},$$

where $\mathbf{1}$ denotes a white reference image, $\mathbf{0}$ an all-zero mask, and $\mathbf{X}^{\text{bg}}$ the background frames, encouraging the model to inpaint the background cleanly.

**Auxiliary Task 2: Reference Inpainting with Camera Control.** This auxiliary task aims to synthesize the reference object onto a background under precise camera control, without requiring removal of any region from the input. Within the unified conditioning interface, we set the source mask to all zeros (indicating no region to remove) and use the background frames as input. The reference image $\mathbf{I}^{\text{ref}}$ is extracted from the first frame of the source input, as in the main task, and the target mask and camera pose guide the object's placement and viewpoint. The conditioning vector is

$$\mathbf{c}_{\text{aux}_2} = \Big( \mathcal{E}_{\text{VAE}}(\mathbf{X}^{\text{bg}}),\ \mathcal{E}_{\text{VAE}}(\mathbf{I}^{\text{ref}}),\ \Pi_s(\mathbf{0}),\ \Pi_s(\hat{\mathbf{M}}^{\text{tgt}}),\ \mathcal{E}_{\text{cam}}(\mathbf{f}) \Big), \qquad \mathbf{X}^{\text{tgt}}_{\text{aux}_2} = \mathbf{X}^{\text{tgt}},$$

As illustrated on the left side of Figure 2, our model complements the multi-task training framework with a two-stage training strategy designed to systematically distribute knowledge across its architecture. This regimen ensures that foundational geometric and pose-related priors are established early, while subsequent refinement focuses on real-world fidelity, thereby balancing generalization and photorealism.

---

[*] This is achieved by setting the normalized device coordinate (NDC) shifts in $\tilde{\mathbf{f}}$ beyond the range $[-1, 1]$, ensuring the projected region exits the frame.

**Stage I: Object Prior & Pose Learning.** In Stage I, we pretrain the model on a large-scale synthetic dataset comprising 3D object meshes rendered under randomized camera poses (including variations in yaw, pitch, distance, and screen-space shifts) against uniform white backgrounds. This controlled environment isolates object appearance from complex scene interactions, allowing the model to focus on learning intrinsic object properties. By jointly optimizing both the main branch and the conditioning control branch, this stage instills category-agnostic object priors and robust camera pose representations, ensuring generalization across diverse objects and viewpoints.

**Stage II: Background Preservation & High-Quality Fine-Tuning.** In Stage II, we fine-tune the model on a curated dataset of high-quality images and videos featuring complex scenes and backgrounds. This stage addresses the limitations of synthetic data by incorporating real-world data, thereby enhancing generalization to natural, intricate environments. To maintain the object priors and pose understanding acquired in Stage I, we freeze the main branch and update only the conditioning control branch, focusing on improved background coherence, and photorealistic rendering.

**Ctrl&Shift** is trained with flow-matching (Lipman et al., 2023). Specifically, let $\mathbf{z}_0 = \mathcal{E}_{\mathrm{VAE}}\big(\mathbf{X}^{\mathrm{tgt}}\big)$ denotes the target latent We sample $t \sim \mathcal{U}(0,1)$ and $\boldsymbol{\varepsilon} \sim \mathcal{N}(\mathbf{0}, \mathbf{I})$ with the same shape as $\mathbf{z}_0$, and define the linear path and target velocity:

$$\mathbf{z}_t \ = \ (1-t)\,\mathbf{z}_0 \ + \ t\,\boldsymbol{\varepsilon}, \qquad \mathbf{v}^\star(\mathbf{z}_t, t) \ = \ \boldsymbol{\varepsilon} - \mathbf{z}_0.$$

A velocity network $\mathbf{v}_\theta(\mathbf{z}_t, \mathbf{c}, t)$ is trained with the flow-matching loss (Lipman et al., 2023):

$$\mathcal{L}_{\mathrm{FM}}(\boldsymbol{\theta}) \ = \ \mathbb{E}_{(\mathbf{z}_0, \mathbf{c})} \, \mathbb{E}_{t \sim \mathcal{U}(0,1), \, \boldsymbol{\varepsilon} \sim \mathcal{N}(\mathbf{0}, \mathbf{I})} \Big[ \| \, \mathbf{v}_{\boldsymbol{\theta}}(\mathbf{z}_t, \mathbf{c}, t) \ - \ \mathbf{v}^\star(\mathbf{z}_t, t) \, \|_2^2 \Big].$$

## 2.4 DATASET CONSTRUCTION

As illustrated in Figure 4, we construct paired examples $\big(\mathbf{X}^{\mathrm{src}}, \mathbf{s}^{\mathrm{src}}\big)$ and $\big(\mathbf{X}^{\mathrm{tgt}}, \mathbf{s}^{\mathrm{tgt}}\big)$. The source pair is a real photograph; the target pair is synthesized by rendering the same object under a novel camera and harmonizing it with the source background.

**Object mesh reconstruction.** Given a source image $\mathbf{I}^{\mathrm{src}}$ (extract from first frame of $\mathbf{X}^{\mathrm{src}}$) and its object mask $M^{\mathrm{src}}$, we reconstruct a watertight, textured mesh $\mathcal{M}$ of the foreground object using Hunyuan3D-2 (Zhao et al., 2025).

**Source camera estimation.** We recover $\mathbf{s}^{\mathrm{src}}$ by aligning $\mathcal{M}$ to the observed silhouette via differentiable rendering. Let $\mathcal{R}(\mathcal{M}, \mathbf{s})$ denote a renderer producing a soft silhouette. We solve

$$\mathbf{s}^{\mathrm{src}} = \arg \max_{\mathbf{s}} \mathrm{IoU}\big(\mathcal{R}(\mathcal{M}, \mathbf{s}), \, \mathbf{M}^{\mathrm{src}}\big),$$

implemented with gradient-based optimization (Chen et al., 2021; Jatavallabhula et al., 2019; Laine et al., 2020), and retain only instances with $\mathrm{IoU} \geq 0.90$. This yields reliable source-camera poses.

**Target camera sampling and object rendering.** We obtain a novel target view $\mathbf{s}^{\mathrm{tgt}}$ by sampling $(\mathrm{yaw}, \mathrm{pitch}, d, r_x, r_y)^\top$ as moderate perturbations of $\mathbf{s}^{\mathrm{src}}$. We then render the object $\mathbf{O}^{\mathrm{tgt}} = \mathcal{R}_{\mathrm{RGB}}(\mathcal{M}, \mathbf{s}^{\mathrm{tgt}})$, producing a view-consistent foreground under the desired camera.

**Background acquisition and harmonization (object pasting).** We first remove the source object with MiniMax-Remover (Zi et al., 2025a) to obtain a clean background plate $\mathbf{B}$. To seamlessly compose $\mathbf{O}^{\mathrm{tgt}}$ into the scene, we train an object–pasting (harmonization) network $\mathcal{H}$ (See Appendix. A.3) in a self-reconstruction regime: given $\big(\mathbf{B}, \mathbf{M}^{\mathrm{src}}, \mathbf{O}^{\mathrm{src}}\big)$, where $\mathbf{O}^{\mathrm{src}} = \mathcal{R}_{\mathrm{RGB}}(\mathcal{M}, \hat{\mathbf{s}}^{\mathrm{src}})$, the network is supervised to predict $\mathbf{X}^{\mathrm{src}}$. This teaches $\mathcal{H}$ to color-match, relight, and blend object boundaries. During inference, we provide $\big(\mathbf{B}, \hat{\mathbf{M}}^{\mathrm{tgt}}, \mathbf{O}^{\mathrm{tgt}}\big)$ to obtain $\mathbf{X}^{\mathrm{tgt}} = \big(\mathbf{B}, \hat{\mathbf{M}}^{\mathrm{tgt}}, \mathbf{O}^{\mathrm{tgt}}\big)$.

## 3 EXPERIMENTS

Our model extends the Wan-1.3B backbone (Wan et al., 2025) with eight control blocks in the conditioning branch. Training uses a spatial resolution of $640 \times 960$, with video inputs at $61 \times 640 \times 960$. In Stage I, we pretrain for 50k steps using AdamW (learning rate $5 \times 10^{-5}$) and a One-Cycle scheduler on a synthetic dataset of $\sim$2M image pairs, generated by rendering 3D object

meshes with randomized camera poses. In Stage II, we fine-tune for 5k steps on a curated dataset of 100K high-quality image and video pairs. For multi-task training, we balance the main task, auxiliary task 1, and auxiliary task 2 with a weighting of 8:1:1. See Appendix A.2 for details.

**Quantitative Evaluation on ObjectMover-A.** As shown in Table 1, we conduct a zero-shot evaluation on the ObjectMover-A benchmark (Yu et al., 2025). Following Yu et al. (2025), we assess frame-level performance by calculating the PSNR between the target frame and the edited frame. For object identity preservation, we crop out the object and compute the DINO score (Caron et al., 2021), CLIP similarity (Radford et al., 2021), and DreamSim (Fu et al., 2023). As shown in Table 1, our method outperforms existing approaches by a large margin, demonstrating its superior performance on object translation.

**Quantitative Evaluation on GeoEditBench.** To evaluate our approach for object manipulation in real-world images, we developed GeoEditBench, a new dataset with 346 carefully curated image pairs captured by skilled photographers under controlled settings (See Appendix A.5 for details). We report PSNR, DINO score, CLIP score, and DreamSim metrics to gauge background preservation and object identity fidelity. To assess camera pose accuracy, we estimated poses from edited out-

Table 1: Zero-shot Evaluation ObjectMover-A

| Method | PSNR ↑ | DINO ↑ | CLIP ↑ | DreamSim ↓ |
|---|---|---|---|---|
| Drag-Anything | 16.36 | 55.56 | 84.44 | 0.411 |
| 3DiT | 19.72 | 45.30 | 81.69 | 0.514 |
| Paint-by-Example | 20.83 | 55.46 | 85.23 | 0.420 |
| Anydoor | 21.86 | 69.32 | 88.95 | 0.289 |
| MagicFixup | 23.82 | 78.49 | 91.06 | 0.198 |
| ObjectMover | 25.27 | 85.07 | 93.16 | 0.142 |
| **Ours** | **28.69** | **88.07** | **93.58** | **0.075** |

puts and calculated (i) the mean absolute percentage error (MAPE) across pose parameters $(\mathrm{yaw}, \mathrm{pitch}, d, r_x, r_y)$, and (ii) the IoU between the edited object's silhouette and the ground-truth mask. Lower MAPE and higher IoU reflect precise camera control. We compare our method against several baselines, including Drag-Anything (Wu et al., 2024), 3DiT (Michel et al., 2023), VACE (Jiang et al., 2025), Flux-Kontext (Labs et al., 2025), Qwen-Image-Edit (Wu et al., 2025), and Nano-Banana (noa) (implementation details in Appendix A.4), as presented in Table 2. Among these, Drag-Anything, VACE, and Nano-Banana utilize spatial signals (e.g., trajectories, bounding boxes, or target masks) as input, enabling precise object placement and resulting in relatively high DINO, CLIP, and DreamSim scores, as well as improved object IoU. However, these methods exhibit higher pose MAPE, reflecting limited control over object pose. In contrast, methods relying solely on language and coordinate inputs, such as 3DiT and Flux-Kontext, struggle to achieve comparable spatial accuracy. Our approach achieves superior PSNR, indicating excellent background preservation, alongside high DINO, CLIP, and DreamSim scores, demonstrating robust retention of object identity. Furthermore, it attains the lowest pose MAPE and highest object IoU, underscoring its precision in both camera control and object manipulation. Despite these advancements, the results highlight that precise object manipulation in real-world images remains a complex challenge, necessitating further research to enhance performance and robustness across diverse scenarios.

**Qualitative Evaluation.** Figure 5 presents a qualitative comparison of our method against state-of-the-art approaches, including VACE (Jiang et al., 2025), Nano-Banana (noa), Flux-Kontext (Labs et al., 2025), Qwen-Image-Edit (Wu et al., 2025), DragAnything (Wu et al., 2024), and 3DiT (Michel et al., 2023). DragAnything and 3DiT exhibit limited generalization to real-world data, constrained by their reliance on specific training datasets. VACE effectively preserves background consistency but fails to directly manipulate the object, instead shifting the entire frame to indirectly place the object at the target location. Nano-Banana and Qwen-Image-Edit produce high-quality images with consistent backgrounds; however, they struggle to accurately incorporate camera pose changes spec-

Table 2: Zero-Shot Evaluation on GeoEditBench

| Method | Translation | Rotation | PSNR ↑ | DINO-Score ↑ | CLIP-Score ↑ | DreamSim ↓ | Pose MAPE ↓ | Obj IoU ↑ |
|---|---|---|---|---|---|---|---|---|
| Drag-Anything | ✓ | ✗ | 17.65 | 57.24 | 70.85 | 0.205 | 46.81% | 0.56 |
| 3DiT | ✓ | ✓ | 20.56 | 39.16 | 57.76 | 0.280 | 51.62% | 0.39 |
| VACE | ✓ | ✗ | 24.32 | 75.38 | 82.53 | 0.175 | 30.56% | 0.72 |
| Flux-kontext | ✓ | ✓ | 21.57 | 57.97 | 68.35 | 0.229 | 46.76% | 0.47 |
| Qwen-Image-Edit | ✓ | ✓ | 22.72 | 61.62 | 79.77 | 0.221 | 39.56% | 0.52 |
| Nano-Banana | ✓ | ✓ | 26.38 | 78.05 | 85.63 | 0.145 | 24.36% | 0.78 |
| Diffusion Handles | ✓ | ✓ | 24.18 | 72.23 | 82.37 | 0.182 | 36.36% | 0.67 |
| Ours | ✓ | ✓ | **28.71** | **85.23** | **90.44** | **0.112** | **17.70%** | **0.83** |

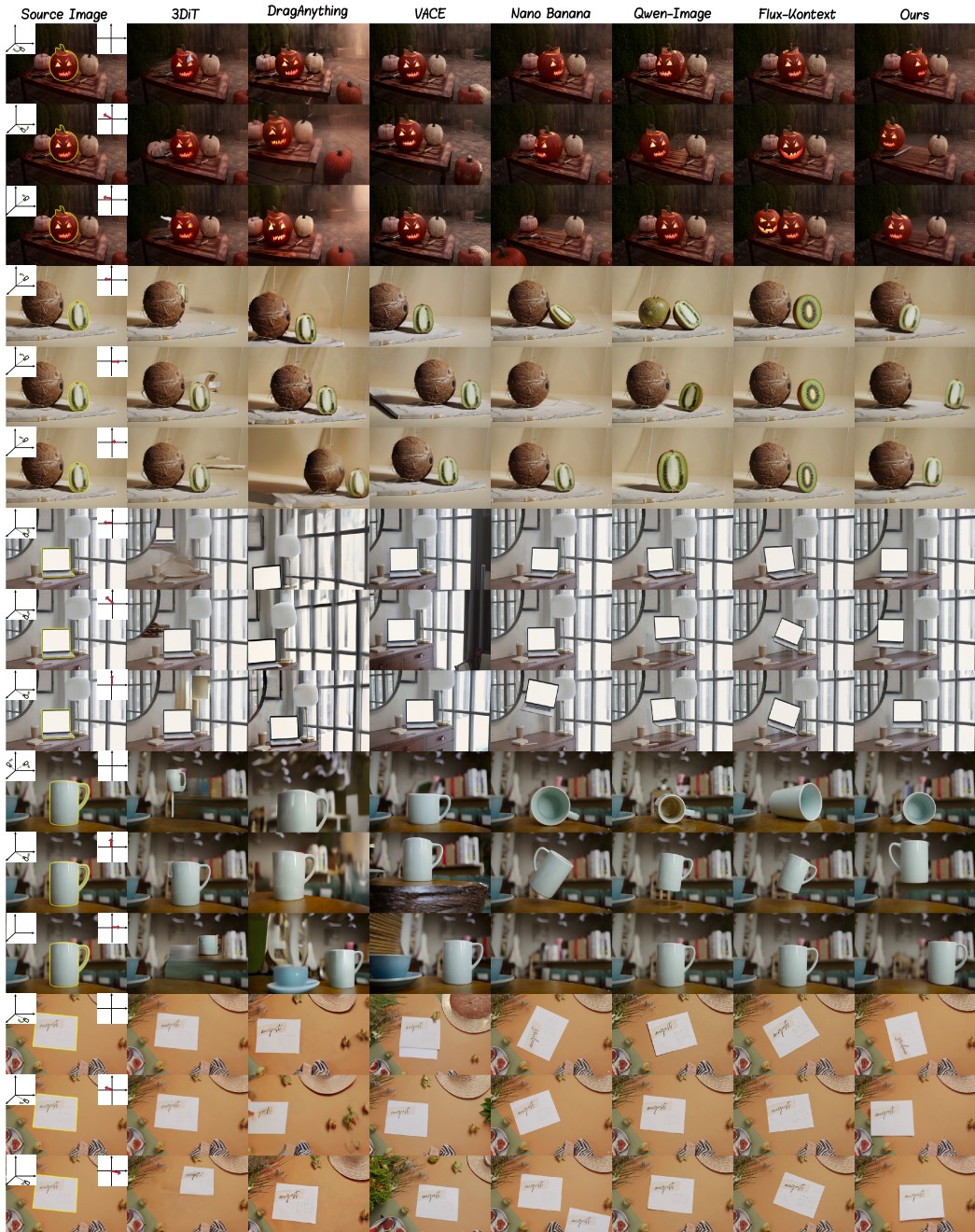

Figure 5: Qualitative comparisons for object manipulation, displaying relative camera changes and NDC shifts. Our model outperforms state-of-the-art methods in background preservation, precise camera pose control, and geometric consistency.

ified via text instructions. In contrast, our method achieves precise camera pose control, enabling accurate object relocation and viewpoint adjustments while maintaining background preservation.

**Ablation Study**: We ablate different stage and task to show the effect of each part of our methods, as shown in 3. Ablation studies on GeoEditBench highlight the contributions of each component in our computer vision pipeline: removing Stage 1 significantly impairs geometric fidelity, with Pose MAPE increase to 32.50% and Obj IoU declining to 0.68, underscoring its role in camera pose understanding. Excluding Stage 2 diminishes reconstruction quality, as evidenced by PSNR dropping to 24.83 and DreamSim rising to 0.195, indicating its importance for background preservation and visual quality. Omitting Aux 1 leads to moderate losses in semantic consistency, with CLIP-Score

falling to 86.32 and DINO-Score to 80.47 alongside a Pose MAPE increase to 23.80%, suggesting it bolsters feature alignment and mid-level visual representations. Dropping Aux 2 most adversely affects object-centric metrics, reducing Obj IoU to 0.65 and elevating Pose MAPE to 28.60% while lowering CLIP-Score to 83.75, revealing its focus on fine-grained supervision for instance-level detection and pose tracking. Overall, the full model achieves superior visual coherence and metric balance, validating the synergistic integration of our multi-task multi-stage training paradigm.

Table 3: Ablation Study on GeoEditBench

| Stage 1 | Stage 2 | Aux 1 | Aux 2 | Main | PSNR ↑ | DINO-Score ↑ | CLIP-Score ↑ | DreamSim ↓ | Pose MAPE ↓ | Obj IoU ↑ |
|---|---|---|---|---|---|---|---|---|---|---|
| ✗ | ✓ | ✓ | ✓ | ✓ | 25.12 | 78.64 | 85.21 | 0.178 | 32.50% | 0.68 |
| ✓ | ✗ | ✓ | ✓ | ✓ | 24.83 | 79.15 | 84.67 | 0.195 | 20.40% | 0.77 |
| ✓ | ✓ | ✗ | ✓ | ✓ | 25.96 | 80.47 | 86.32 | 0.162 | 23.80% | 0.74 |
| ✓ | ✓ | ✓ | ✗ | ✓ | 26.54 | 77.89 | 83.75 | 0.149 | 28.60% | 0.65 |
| ✓ | ✓ | ✓ | ✓ | ✓ | **28.71** | **85.23** | **90.44** | **0.112** | **17.70%** | **0.83** |

## 4 LIMITATIONS AND FUTURE WORK

While our framework achieves precise geometry-aware object manipulation, we acknowledge several limitations stemming from both the data construction pipeline and the inherent complexity of the task (See Appendix A.8 for details). We discuss these challenges and future directions below:

- **From Technical Controllability to Practical Usability.** While our 8D relative pose descriptor $\mathbf{f}$ provides robust geometric control, we recognize that manually specifying these values is not user-friendly. A key future challenge is mapping intuitive user interactions, such as 2D mouse drags, rotational gestures, or 3D gizmo manipulation, directly to this descriptor. Since $\mathbf{f}$ represents a rigid transformation, it can be analytically derived from such inputs, bridging the gap between our precise internal representation and an accessible user experience.

- **Physical Realism Beyond Geometry.** Our framework prioritizes geometric consistency but does not explicitly model physical interactions like lighting, variable shadows, or specular reflections. Instead, we rely on the model to implicitly learn these photometric effects from training data. While often effective, this data-driven approach struggles with physically correct shadows or reflections under drastically different illumination. Integrating explicit lighting estimation or physics-based rendering guidance is a valuable direction to enhance realism.

- **Generalization Boundaries via the Data Pipeline.** Our method's reliance on 3D mesh reconstruction imposes generalization boundaries: **Object Type:** The pipeline assumes rigid geometry, limiting applicability to non-rigid objects (e.g., cloth, hair) or topological changes (e.g., smoke, fluids). **Material Properties:** Transparent or highly reflective objects (e.g., glass, mirrors) are often reconstructed with "baked-in" background textures, causing artifacts during synthesis. **Complex Occlusions:** Our "remove-and-inpaint" strategy assumes the target area is visible or planar and cannot handle complex depth relationships, such as moving an object behind another scene element, which requires reasoning about occluding background geometry.

- **Limited Video Manipulation Capabilities**: While our method achieves precise geometric control, video manipulation shows lower visual fidelity than image editing. This arises from our rigid 3D reconstruction pipeline's inability to capture non-rigid deformations and complex spatiotemporal dynamics in real-world videos. Future work will leverage 4D representations to improve video manipulation quality.

## 5 CONCLUSION

In this work, we introduce **Ctrl&Shift**, a novel diffusion-based framework for precise object-level manipulation in images and videos, enabling relocation and viewpoint control while preserving scene integrity through task decomposition into removal and reference-guided inpainting, multi-task multi-stage training for signal disentanglement, and a curated dataset. Extensive evaluations demonstrate its superior fidelity, temporal coherence, and controllability across real-world scenarios, positioning it as a versatile tool for visual content creation, augmented reality, and film production, with future extensions targeting dynamic scenes and multi-object interactions.

## 6 ACKNOWLEDGMENTS

This work is partially supported by the Hong Kong General Research Fund under Grant PolyU 15205924 and the National Science Foundation of China under Grants 62250710682 and 61761136008. We also acknowledge the support from the Research Institute for Artificial Intelligence of Things at The Hong Kong Polytechnic University.

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

# A APPENDIX

## A.1 LLM USAGE

In preparing this manuscript, we utilized ChatGPT and Grok, solely for language polishing and minor refinements to improve clarity, grammar, and flow in the text. The LLM was provided with sections of the draft and asked to suggest revisions, which were then reviewed, edited, and incorporated by the authors as deemed appropriate. All core ideas, research contributions, technical details, and analyses originate from the authors and were not generated or ideated by the LLM. No other LLMs were used in the research process.

## A.2 TRAINING DETAILS

**Training Details.** We build on the Wan-1.3B backbone with eight control blocks. In *Stage I*, we train for 50k optimizer steps using AdamW (learning rate $5 \times 10^{-5}$) with a One-Cycle scheduler. The training resolution is $640 \times 960$ for images and $61 \times 640 \times 960$ for videos. We use 32 A100 GPUs for training, employing mixed precision with bfloat16 and DeepSpeed zero optimization stage 2. During training, we randomly drop the camera pose condition with a 0.1 probability; during inference, we apply a classifier-free guidance (CFG) scale of 1.5 with UniPC (Zhao et al., 2023) sampler for 40 steps. *Stage II* fine-tuning runs for 5k steps on mixed image/video data), retaining the same optimization setup unless otherwise specified.

**Dataset Construction Details.** We curate a large-scale corpus for object manipulation through a two-stage process. *Source Acquisition and Filtering.* We start with approximately 400k videos from Pexels, using the first frame of each clip as the source image for mesh reconstruction. In parallel, we synthesize ~100k diverse object-centric images using HunyuanT2I. We apply prompt-based filtering to exclude non-manipulable categories (e.g., roads, buildings) and discard instances exhibiting disconnected components, heavy truncation, or obvious deformations. *Mask Extraction and 3D Reconstruction.* For each retained source image, we obtain an object mask using Grounded-SAM-2 (Ren et al., 2024) and reconstruct a textured, watertight mesh with Hunyuan3D-2 (Zhao et al., 2025). This process yields ~100k object meshes after filtering. *Pair Synthesis for Stage I:* For each object mesh, we randomly sample 20 different camera views and object placements to render ~2M image pairs

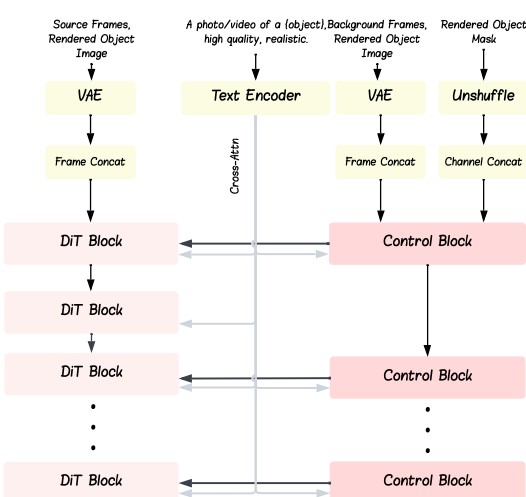

Figure 6: Training of the object pasting model. The model is trained to recover the real-world image/video given background frames and rendered object image.

for pretraining in Stage I. *High-Quality Subset for Stage II:* For Stage II, we utilize only meshes derived from the Pexels dataset. We estimate the source camera pose via differentiable rasterization and filter out images/videos where the source camera estimate is unreliable (IoU $\leq 0.90$). For the retained samples, we render the mesh under a new camera view and employ our object pasting model to composite the rendered object onto the background image/video. We further conduct an automatic harmonization sanity check using ChatGPT-Image-1 to exclude poor pastes. The resulting Stage II dataset comprises roughly ~50K video pairs and ~50K image pairs for high-fidelity fine-tuning.

## A.3 OBJECT PASTING MODEL

As illustrated in Figure 6, our object pasting model adopts an architecture similar to that of the object manipulation network. This model is designed to composite the rendered object image onto the background frames, harmonizing it with the surrounding scene by adjusting lighting, shadows,

and other visual attributes while preserving the object's original orientation and position. The inputs include the background frames, the rendered object image, the corresponding rendered object mask, and a textual prompt. During training, the rendered object image and mask are generated using the estimated source camera pose $\mathbf{s}^{\text{src}}$, with the objective of reconstructing the original source video. This process enables the model to learn precise object pasting and seamless integration with the background. At inference, the rendered object image and mask are provided at the target location within the same background, yielding the target frames $\mathbf{X}^{\text{tgt}}$. We employ Wan-1.3B as the backbone, augmented with 16 control layers; the backbone is frozen, and only the control branch is updated during training. The model is trained for 20K steps at a learning rate of $5 \times 10^{-5}$ on 16 A100 GPUs. During inference, we apply cfg of 5.0 with 40 steps using UniPC sampler.

## A.4 BASELINE IMPLEMENTATION DETAILS

**Implementation Details of 3DiT.** We use the pretrained checkpoint of 3DiT, we first translate the object according to our NDC shifts (convert from [-1,1] to [0,1] to match the convention of 3DiT). For rotation, since 3DiT accepts the object rotation angle as input, while our $\mathbf{R}_{\text{rel}}$ represents the relative camera rotation, we extract the angle as the negative of the yaw from $\mathbf{R}_{\text{rel}}$:

$$\theta = -\text{atantwo}(\mathbf{R}_{\text{rel}}[0, 2], \mathbf{R}_{\text{rel}}[0, 0]),$$

which accounts for the opposite direction between camera and object rotations. We adopt the default settings of cfg 3.0 and the input image is resized to $256 \times 256$ to match 3DiT's resolution and resized back after generation for qualitative and quantitative comparisons.

**Implementation Details for DragAnything.** We employ the pretrained checkpoint of DragAnything and draw a linear Gaussian path starting from the source NDC coordinates and ending at the target NDC coordinates. The input is resized to $320 \times 576$ in accordance with its default settings. We adopt the default settings, with CFG of 3.0, and generate 25 frames video. We then extract the last frame as the edited result for comparison.

**Implementation Details for VACE.** We utilize the pretrained VACE-1.3B model, adapting its "moving anything" configuration to our setting. Specifically, we compute the bounding boxes for the source and target masks, convert them into video-like signals for input to the VACE control branch, and guide generation using the prompt template: "A {obj} is moving to the {top/bottom}, {left/right} side of the image, with terminal location {rx}, {ry}." We adopt the default settings to generate videos at a resolution of $81 \times 480 \times 832$ with a CFG scale of 5.0. The last frame from the generated video is extracted for comparison.

**Implementation Details of Qwen-Image.** We utilize the pretrained checkpoint of Qwen-Image-Edit. The model takes the source image as input, guided by a refined prompt template: "Move the {object} to the {top/bottom}, {left/right} side of the image at target location {$r_x$}, {$r_y$}. Rotate the object horizontally by {yaw$^{\text{tgt}}$ - yaw$^{\text{src}}$} degrees and vertically by {pitch$^{\text{tgt}}$ - pitch$^{\text{src}}$} degrees while preserving the background and other objects unchanged." We employ the default settings, with a classifier-free guidance scale of 4.0, 50 inference steps, and a resolution of $832 \times 1248$.

**Implementation Details of Flux-Kontext.** We utilize the official pretrained checkpoint, FLUX.1-Kontext-dev for inference. The model takes the source image as input, guided by a refined prompt template: "Move the {object} to the {top/bottom}, {left/right} side of the image at target location {$r_x$}, {$r_y$}. Rotate the object horizontally by {yaw$^{\text{tgt}}$ - yaw$^{\text{src}}$} degrees and vertically by {pitch$^{\text{tgt}}$ - pitch$^{\text{src}}$} degrees while preserving the background and other objects unchanged." We employ the default settings, with a classifier-free guidance scale of 5.0, 20 inference steps, and a resolution of $960 \times 640$.

**Implementation Details of Nano-Banana.** We use the official GEMINI-2.5-FLASH-IMAGE-PREVIEW API to generate edits from a source image $\mathbf{I}^{\text{src}}$ and an estimated target mask $\hat{\mathbf{M}}^{\text{tgt}}$. The mask $\hat{\mathbf{M}}^{\text{tgt}}$ is obtained by taking the minimum enclosing square of the source mask $\mathbf{M}^{\text{src}}$ and translating it to the target position using normalized device–coordinate (NDC) shifts $(r_x, r_y)$ (cf. Sec. 2.1). We provide the API with $\mathbf{I}^{\text{src}}$, $\hat{\mathbf{M}}^{\text{tgt}}$, and the following structured prompt: "Move the {object} to the {top/bottom}, {left/right} side of the image at target location {$r_x$}, {$r_y$} so that its center lies inside the white square mask. Rotate the object by {yaw$^{\text{tgt}} -$ yaw$^{\text{src}}$} degree and {pitch$^{\text{tgt}} -$ pitch$^{\text{src}}$} degree, while preserving the background and all other objects unchanged."

### A.5 DETAILS OF THE GEOEDITBENCH

To evaluate the performance of our object manipulation framework, we manually curated the GeoEd-itBench dataset. Specifically, we collected 20 common objects and, for each, randomly combined it with two or three other objects, then captured images with the target object moved and rotated across 5 different locations and viewpoints. This process was repeated 3 times per object, yielding 300 source images for the benchmark. Additionally, we captured 4 multi-view images (front, back, left, and right) of each isolated object and reconstructed high-quality 3D meshes using the Hunyuan3D API. We manually filtered out meshes that did not accurately conform to the object images and annotated object masks using Grounded-SAM-2. Camera poses were estimated via differentiable rendering, and to establish a reliable benchmark, we retained only images where the rendered silhouette achieved an IoU $\geq 0.95$ with the annotated mask. After filtering, the dataset comprises 346 high-quality image pairs. To promote reproducibility and community value, we will open-source GeoEditBench.

### A.6 FAILURE CASE ANALYSIS

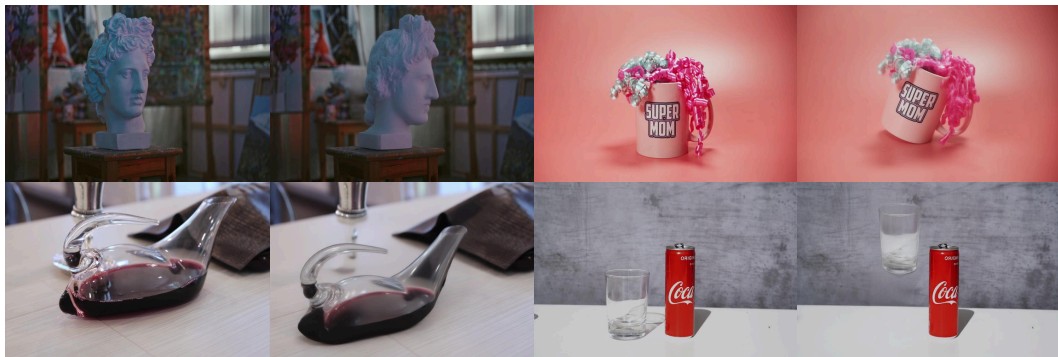

Figure 7: **Limitations of Ctrl&Shift.** Left: source, Right: edited result. Top: Loss of high-frequency details in objects with complex textures. Bottom: Artifacts when handling transparent objects due to texture-baking errors in the 3D reconstruction pipeline.

Figure 7 illustrates current limitations of **Ctrl&Shift**. In the first row, we observe a loss of fidelity when editing objects with highly complex, fine-grained textures. This issue stems from the synthetic data generation pipeline; specifically, the image-to-mesh reconstruction step often fails to capture high-frequency geometric and texture details. Consequently, the model is trained on synthetic pairs that lack this granularity, limiting its ability to preserve intricate patterns during inference. In the second row, the model struggles with transparent objects. This limitation also originates from the underlying 3D reconstruction method, which cannot explicitly model transparency or refraction. Instead, it erroneously "bakes" background information onto the object's surface texture. This error propagates to the editing model, causing the generated output to appear blurry or opaque, as the model cannot correctly synthesize the dynamic background changes required for transparent materials in a new pose.

## A.7 MULTI-OBJECT EDITING

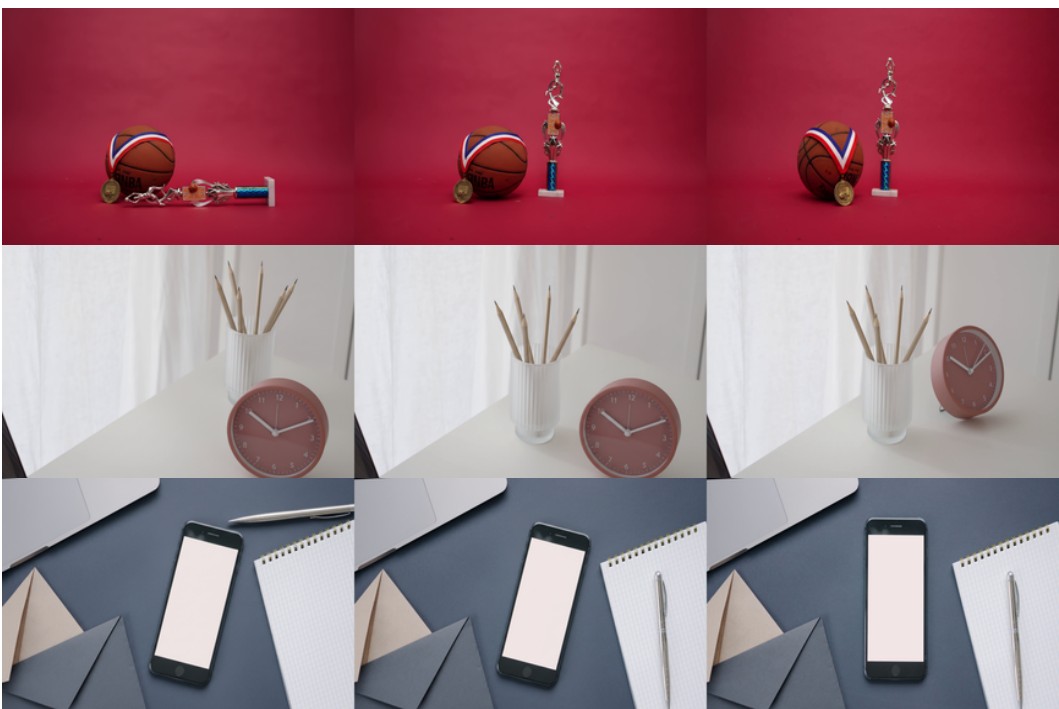

Figure 8: Multi-object editing results achieved through sequential editing of distinct objects. From left to right: source image, first edit, second edit.

As shown in Figure 8, we present the results of multi-object editing using our method, which is achieved through sequential editing of distinct objects. Our framework does not currently support simultaneous multi-object editing, and we believe this represents a promising direction for future work.

## A.8 DETAILED DATA PREPROCESSING PIPELINE AND STATISTICS

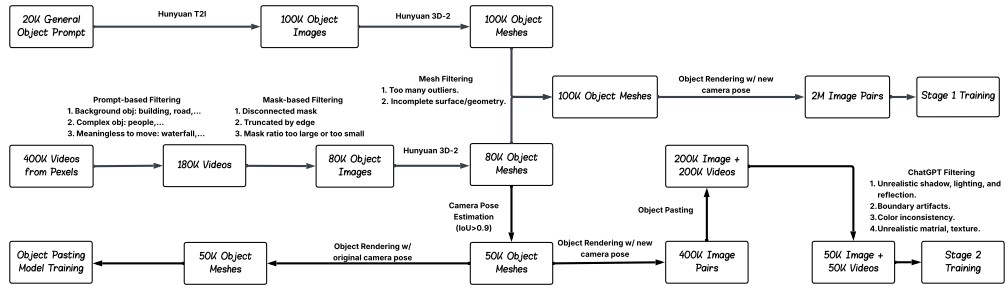

Figure 9: Detailed data preprocessing pipeline and statistics.

Our data pipeline is designed to overcome the scarcity of high-quality paired data for object manipulation. It operates in two distinct phases: *Stage 1*, which utilizes large-scale synthetic data to establish fundamental geometric consistency, and *Stage 2*, which leverages curated real-world video data to enforce photorealism and correct texture handling.

### A.8.1 LARGE-SCALE SYNTHETIC DATA GENERATION

The primary objective of this stage is to create a massive volume of object pairs with perfect geometric ground truth to train the model's manipulation capabilities. We begin by collecting a diverse set of 20k general object prompts, which are processed by Hunyuan T2I to generate 100k synthetic object images ensuring a wide variety of categories and appearances. These 2D images are lifted to 3D using Hunyuan 3D-2. To guarantee geometric quality, we apply a mesh filtering step that identifies and discards meshes exhibiting excessive outliers or incomplete surfaces, resulting in a clean set of 100k verified object meshes. Finally, we render these meshes from randomized novel camera poses; by pairing the original views with these novel views, we generate a large-scale dataset of 2 million image pairs.

### A.8.2 REAL-WORLD DATA SYNTHESIS

To bridge the domain gap between synthetic renderings and real-world imagery, we curate a high-fidelity dataset starting from a raw collection of 400k videos sourced from Pexels.

**Data Proprecessing and Filtering.** We first employ prompt-based semantic filtering to exclude subjects unsuitable for rigid manipulation, including background structures (e.g., buildings, roads), deformable entities (e.g., people), and amorphous textures (e.g., waterfalls); this reduces the corpus to 180k videos. Subsequently, we apply mask-based geometric filtering to ensure high-quality segmentation, pruning instances that exhibit disconnected masks, frame-edge truncation, or extreme size ratios. This rigorous filtering process yields a set of 80k candidate object images.

**3D Validation and Alignment.** The candidate images are processed through Hunyuan 3D-2 to generate corresponding 3D meshes. Since single-view reconstruction from real images is prone to misalignment, we enforce a strict consistency check. We perform camera pose estimation to reproject the generated mesh back onto the 2D image plane and calculate the Intersection over Union (IoU) between the projected mesh mask and the original object mask. Only meshes satisfying an $IoU > 0.9$ are retained, resulting in a high-quality subset of 50k verified object meshes.

### A.9 ABLATION STUDY ON TARGET MASK STRATEGY

Table 4: Ablation Study on Target Mask Strategy

| Training | Inference | PSNR ↑ | DINO ↑ | CLIP ↑ | DreamSim ↓ | Pose MAPE ↓ | Obj IoU ↑ |
|---|---|---|---|---|---|---|---|
| Estimated | Estimated | 28.71 | 85.23 | 90.44 | 0.112 | 17.70% | 0.83 |
| Estimated | GT | 28.95 | 85.51 | 90.48 | 0.110 | 17.35% | 0.84 |
| GT | GT | 29.58 | 86.42 | 90.89 | 0.104 | 16.20% | 0.86 |
| GT | Estimated | 26.83 | 80.67 | 84.72 | 0.185 | 25.85% | 0.71 |

To demonstrate the robustness of our mask estimation strategy, we present an ablation study in Table 4. The results reveal several key insights: First, when training with estimated masks, providing ground-truth masks at inference yields only marginal improvement (Row 2 vs. Row 1), as the model has learned to rely primarily on the pose descriptor rather than precise mask boundaries for geometric manipulation. That means our current deisgn is not sensitive to the target mask estimation. Second, training and inference with ground-truth masks achieves the best performance (Row 3), as accurate masks provide richer geometric information that simplifies the learning task. However, this approach is impractical for real-world deployment where ground-truth masks are unavailable without explicit 3D reconstruction. Most importantly, Row 4 demonstrates that models trained on ground-truth masks suffer significant performance degradation when only estimated masks are available at inference, confirming that such models develop a strong dependency on precise mask information. In contrast, our approach (Row 1) maintains robust performance under realistic conditions, validating our design choice to train with estimated masks for practical applicability.

### A.10 ABLATION STUDY ON STAGE II TRAINING STRATEGY

Freezing the backbone during Stage II prevents the model from overfitting to the narrower Stage II distribution and catastrophically forgetting the broad geometric understanding acquired in Stage

I. To validate this design choice, we conducted an ablation study comparing our approach (frozen main branch with trainable control branch) against fully fine-tuning both branches in Stage II, as shown in Table 5. The results demonstrate that while full fine-tuning achieves marginally better photometric quality (PSNR, DreamSim), it significantly degrades geometric accuracy (Pose MAPE: 17.70% vs. 21.45%) and object boundary preservation (Obj IoU: 0.83 vs. 0.79). This confirms our hypothesis that unrestricted fine-tuning on the smaller Stage II dataset causes the model to lose its robust geometric manipulation capabilities. Our frozen-backbone approach strikes a better balance, maintaining strong geometric control while still learning effective photometric harmonization through the trainable control branch.

Table 5: Ablation Study on Stage II Training Strategy (Zero-shot Evaluation on GeoEditBench)

| Stage II Strategy | PSNR ↑ | DINO ↑ | CLIP ↑ | DreamSim ↓ | Pose MAPE ↓ | Obj IoU ↑ |
|---|---|---|---|---|---|---|
| Freeze main, train control | 28.71 | 85.23 | 90.44 | 0.112 | 17.70% | 0.83 |
| Fine-tune main & control | 29.12 | 84.85 | 90.38 | 0.108 | 21.45% | 0.79 |

### A.11 MORE RESULTS

| Source Image | Edited Result | Source Image | Edited Result |
|:---:|:---:|:---:|:---:|

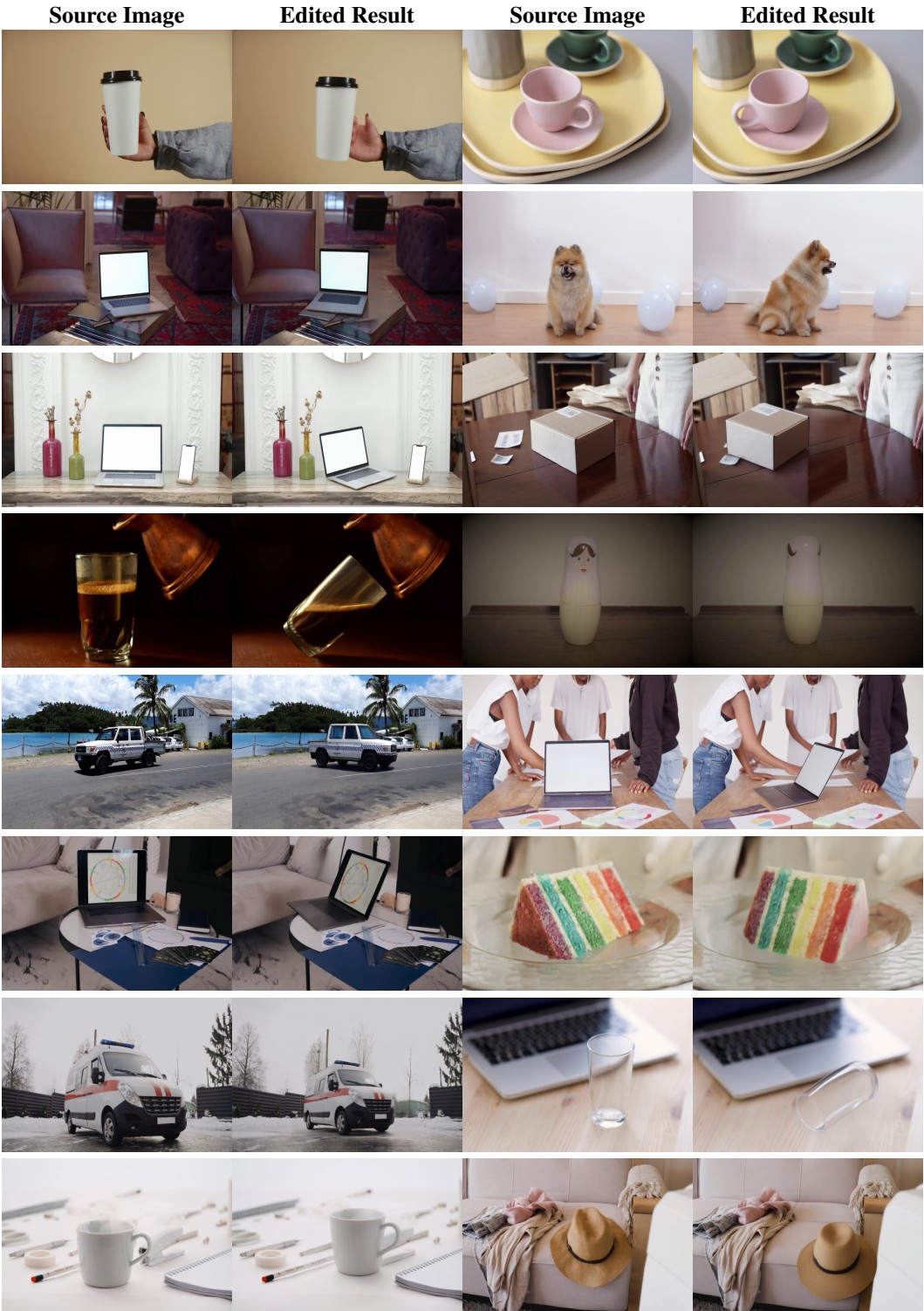

Figure 10: Additional Results on Object Manipulation.

| Source Image | Edited Result | Source Image | Edited Result |
|---|---|---|---|

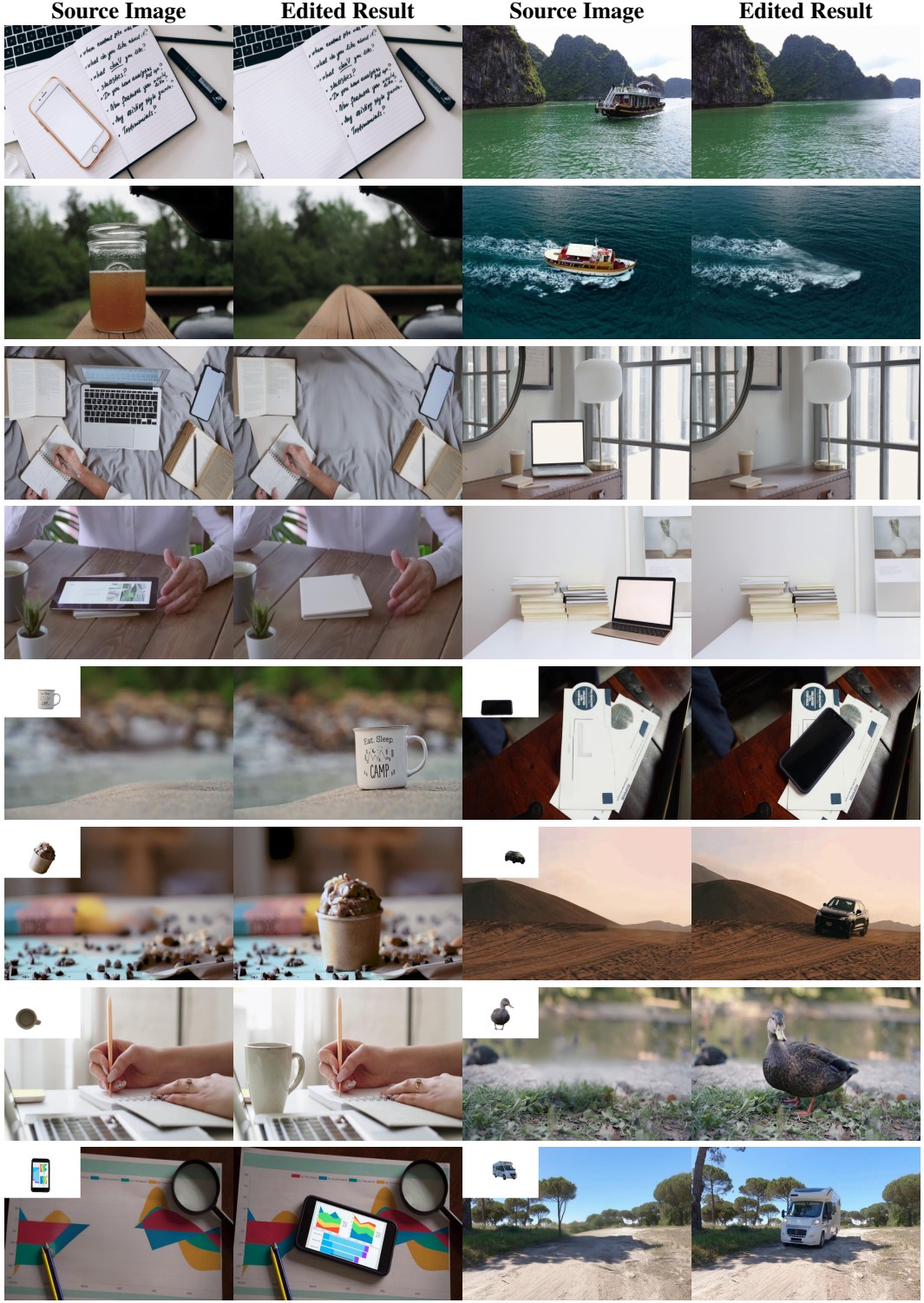

Figure 11: Additional Results on Object Removal and Reference Image Inpainting with Camera Control.

