# OpenReview forum: "CTRL&SHIFT: High-quality Geometry-Aware Object Manipulation in Visual Generation"
_ICLR.cc/2026/Conference — ICLR 2026 Poster_

### Official Review · Reviewer_sMZf · 2025-10-16

**Soundness:** 3
**Presentation:** 1
**Contribution:** 2
**Rating:** 4
**Confidence:** 3

**Summary:**

This paper presents CTRL&SHIFT, a 3D-aware image editing approach. Given an image, the target location (specified by binary mask), and a relative camera pose transformation, the method is able to translate and rotate the object to the desired pose. The core model architecture is similar to VACE, with Control Blocks (Context Blocks) injecting the control signals. To train the model, authors design an automatic data construction pipeline, which produces paired images and ground-truth transformation. The training is done in two stages: first learning object-centric novel view synthesis, then learning blending objects to real-world background scenes. Quantative results on two benchmarks show clear improvement of CTRL&SHIFT over previous baselines.

**Strengths:**

- While image editing is a well-studied task, 3D-aware editing is underexplored. This work brings recent advances in general editing to the field, and proposes a large-scale dataset to further improve results.
- The background preservation when editing foreground objects seems to be pretty good. The visual effects like shadows also look realistic.
- The new GeoEditBench can be a useful benchmark to the community.

**Weaknesses:**

1. The paper presentation needs a lot of improvement. The main paper spends too much space talking about details of the data format (Sec. 2.1, 2.2, 2.3), yet the pipeline figure is in the Appendix. I'd suggest introducing the model architecture first, and then define each input & output and their shapes;
2. Similarly, the main paper only has one paragraph (in Sec.1) discussing related works. This makes it hard to position the paper in the literature. Please move the Related Work part from Appendix to the main paper.
3. Methodology-wise, while authors claim that architectural design is a contribution of the paper, I don't see why this is novel compared to ObjectMover and VACE. ObjectMover first shows that one can leverage video priors for more consistent object editing. VACE proposes the Context Block control mechanism, which is similar to the Control Blcok in this work. The Reference Images, Source / Target Masks, are conceptually similar to the Context Tokens in VACE. The only difference seems to be the injection of camera pose, yet it has been extensively studied in novel view synthesis and camera-controlled video generation papers.
4. I appreciate the effort to compare with 6 baselines. However, why not compare with methods that explicitly use 3D bounding boxes or 3D representations such as Diffusion Handles and Image Sculpting? An even simpler baseline can be simply conditioning the diffusion model on the input frames, the reference image, and the 3D object bounding box in the target frame (projected with MLP similar to how you handle camera poses).

**Questions:**

1. For camera conditioning, have you tried common methods in camera-controlled generation, such as Plucker embedding?
2. I'm also a bit confused by the model architecture. In the dataset construction pipeline, it seems that each data sample contains only 2 images (source and target). However, I believe Wan's video VAE works for `1+4*n` frames. How do you get other frames as model input here?

---

> ### Author Response · Authors · 2025-11-21
> **Response to Reviewer sMZf**
>
> # Response to Reviewer sMZf
>
> We sincerely thank you for the constructive feedback and valuable suggestions. We address your concerns point by point below:
>
> ## 1. Paper presentation and organization:
>
> We greatly appreciate your suggestions regarding paper structure. Following your recommendations, we have reorganized the revised manuscript to:
>
> - Move the model architecture (previously in Appendix) to the main paper, presenting it before the detailed data format descriptions
> - Relocate the Related Work section from the Appendix to Section 2 of the main paper, providing proper contextualization within the literature
>
> We believe these changes significantly improve the paper's readability and positioning within the field.
>
> ## 2. Clarification on architectural design and novelty:
>
> We sincerely appreciate your careful analysis of our work in relation to ObjectMover and VACE. We acknowledge that our work builds upon these important prior contributions, and we apologize if our claims about "architectural design" as a contribution were unclear or overstated.
>
> **Clarification on "architectural design" contribution:** We acknowledge that the individual architectural components we use (Control Blocks, Camera Pose Encoding) are well-established in the literature. Our use of "architectural design" was intended to refer to the **systematic framework design and training strategy**, not the invention of novel architectural primitives. Specifically, our contributions lie in:
>
> - **Multi-task training design**: Our deliberate decomposition into main manipulation, object removal, and object inpainting tasks (Section 2.4) to disentangle background preservation, object identity, spatial mask and geometric transformation, which is validated through ablation studies (Table 3).
>
> - **Two-stage training strategy**: The systematic approach of first learning geometric priors on large-scale synthetic data (Stage I) before adapting to real-world photometric properties (Stage II), specifically designed to address the data scarcity and geometric accuracy requirements of our task.
>
> - **Pose descriptor design**: While camera conditioning has been studied in novel view synthesis, those methods change the viewpoint of the **entire scene**. Our 8D descriptor is specifically designed for **object-only** manipulation while preserving background and whole scene camera viewpoint, a fundamentally different problem requiring different design choices.
>
> As stated in our contribution part: **"Architectural Design. We disentangle background, object identity, spatial masks, and geometric transformations through a multi-task, multi-stage training strategy that mirrors the semantic structure of object manipulation."** This systematic decomposition and training strategy, rather than novel architectural primitives, constitutes our architectural contribution.
>
> ### Distinction from ObjectMover:
> While ObjectMover demonstrates the value of leveraging video priors for object editing, there are fundamental differences in task scope and approach:
>
> - **Task complexity**: ObjectMover primarily addresses object **shifting**, essentially 2D translation, which is substantially simpler than our full 8-DOF geometric manipulation. Collecting paired data for simple shifting is feasible from real videos (tracking the same object across frames), whereas paired data for arbitrary geometric transformations with ground-truth pose annotations is extremely scarce in real-world videos. This data scarcity motivates our synthetic data generation strategy in Stage I.
>
> - **Image vs. video generalization**: ObjectMover is designed specifically for image editing and cannot directly handle video manipulation, as it relies on temporal consistency cues from video sequences. Our framework is designed to handle both images and videos uniformly through our pose-conditioned control mechanism, providing greater flexibility for practical applications.
>
> We have revised the paper to more clearly position these contributions and avoid overstating the novelty of individual architectural components. Thank you for helping us clarify this important distinction.

---

> > ### Author Response · Authors · 2025-11-21
> > **Response to Reviewer sMZf**
> >
> > ## 3. Comparison with 3D-based methods (Diffusion Handles, Image Sculpting):
> >
> > This is an excellent suggestion. We acknowledge that comparing with methods that explicitly use 3D representations would strengthen our evaluation. To compare with Diffusion Handles, we present the following results,
> >
> > **Table: Zero-Shot Evaluation on GeoEditBench**
> >
> > | Method | Translation | Rotation | PSNR ↑ | DINO-Score ↑ | CLIP-Score ↑ | DreamSim ↓ | Pose MAPE ↓ | Obj IoU ↑ |
> > |--------|-------------|----------|--------|--------------|--------------|------------|-------------|-----------|
> > | Drag-Anything | ✓ | ✗ | 17.65 | 57.24 | 70.85 | 0.205 | 46.81% | 0.56 |
> > | 3DiT | ✓ | ✓ | 20.56 | 39.16 | 57.76 | 0.280 | 51.62% | 0.39 |
> > | VACE | ✓ | ✗ | 24.32 | 75.38 | 82.53 | 0.175 | 30.56% | 0.72 |
> > | Flux-kontext | ✓ | ✓ | 21.57 | 57.97 | 68.35 | 0.229 | 46.76% | 0.47 |
> > | Qwen-Image-Edit | ✓ | ✓ | 22.72 | 61.62 | 79.77 | 0.221 | 39.56% | 0.52 |
> > | Nano-Banana | ✓ | ✓ | 26.38 | 78.05 | 85.63 | 0.145 | 24.36% | 0.78 |
> > | Diffusion Handles | ✓ | ✓ | 24.18 | 72.23 | 82.37 | 0.182 | 36.36% | 0.67 |
> > | **Ours** | ✓ | ✓ | **28.71** | **85.23** | **90.44** | **0.112** | **17.70%** | **0.83** |
> >
> >
> > **Regarding Image Sculpting**, we note that this method operates under a significantly different computational paradigm, **requiring explicit 3D reconstruction and optimization in NeRF space for each image.** This diverges fundamentally from the primary objective of our work: to enable efficient, zero-shot editing without test-time optimization or explicit 3D reconstruction. Given these differences in inference scope and computational cost, we respectfully submit that Image Sculpting is not a suitable baseline for direct comparison in this setting.
> >
> > ## 4. Why not use Plücker embeddings for camera conditioning?
> >
> > While Plücker embeddings are highly effective for global camera control (e.g., novel view synthesis), they are fundamentally unsuitable for our local object manipulation task:
> >
> > **Invariance of Global Camera Pose.** Plücker embeddings encode camera rays for the entire image plane to facilitate global viewpoint changes. In our setting, the scene camera remains static; Consequently, the global Plücker embedding would remain identical between source and target, failing to provide any control signal for the object's local rotation and translation.
> >
> > **Causal Dependency at Inference for Local Camera Pose of Object.** Plücker embeddings are spatially dense representations that associate specific rays with specific pixels. To construct a valid Plücker map for **the only the manipulated object**, we would need to know the precise pixel occupancy (mask) of the target object before obtaining the Plücker embeddings. This creates a circular dependency: we cannot compute the correct per-pixel rays without the target mask, yet we cannot predict the target mask without first generating the image.
> >
> > **Efficiency**: Our 8D pose descriptor is a lightweight, object-centric vector that abstracts away specific pixel coordinates. This allows us to control geometry effectively without the heavy computational overhead of processing high-resolution, dense ray maps.
> >
> > ## 5. Clarification on Model Architecture and Video Input
> >
> > We apologize for the confusion regarding the model architecture and data flow. We have revised Figure 4 to provide better clarity on this aspect.
> >
> > - **Flexible input handling:** Both our object pasting model and object editing model are designed to handle both image and video inputs, not just image pairs.
> >
> > - **Processing pipeline for videos:** For video data, we perform image-to-mesh reconstruction and camera pose estimation on the first frame only. The reconstructed 3D object is then rendered to generate a reference object image, **which is fed to our object pasting model along with the background video. The model performs reference-based inpainting, pasting the rendered object into the background video while maintaining temporal consistency.** This process generates our paired training video.
> >
> > We hope this clarifies the architecture and data flow in our pipeline.

---

> > > ### Comment · Reviewer_sMZf · 2025-11-22
> > >
> > > I thank the authors for the rebuttal. The new paper layout now looks much better. The comparison with Diffusion Handles also makes the benchmark stronger. I agree that Image Sculpting is under a different setting. Finally, I agree with the authors that the entire framework is indeed a contribution. Please revise the claim in the paper to make it more clear.
> > >
> > > I will raise my score to 6.

---

> > > > ### Author Response · Authors · 2025-11-22
> > > > **Response to Reviewer sMZf**
> > > >
> > > > Thank you so much for taking the time to reconsider our work and for your continued feedback. We really appreciate your acknowledgment of the improvements we've made. We'll make sure to clarify our contributions more explicitly in the revision as you recommended. Thanks again for all your effort and valuable input!

---

### Official Review · Reviewer_19Wi · 2025-10-29

**Soundness:** 3
**Presentation:** 3
**Contribution:** 3
**Rating:** 8
**Confidence:** 3

**Summary:**

This paper introduces Ctrl&Shift, a novel diffusion framework for high-quality, geometry-aware object manipulation in images and videos. The core challenge it addresses is the fusion of precise controllability from geometry-based methods with the realism and generalization of diffusion-based approaches.

The paper's main contribution is to cleverly decompose the complex manipulation task into two sub-tasks: 1) object removal and 2) reference-guided inpainting under explicit camera pose control. The authors design an 8-dimensional relative camera pose descriptor, f, which is injected as a control signal into the diffusion model. This enables end-to-end geometric control during inference without requiring explicit 3D modeling.

To achieve this, the paper proposes a multi-task, multi-stage training strategy, supplemented by a sophisticated data construction pipeline to generate paired, geometry-supervised, real-world data. Experimental results on the authors' new GeoEditBench and the ObjectMover-A benchmark demonstrate state-of-the-art performance, outperforming existing methods in fidelity, identity preservation, and geometric control accuracy (e.g., pose MAPE and IoU).

**Strengths:**

1. Conceptual Innovation: The paper's primary strength lies in its core idea. It represents a conceptual shift: instead of relying on expensive or unstable 3D reconstruction (like NeRF or Mesh) at inference time, it injects precise geometric control (a relative pose vector) as a condition into the 2D diffusion process. This is a very clever decoupling that elegantly combines the advantages of both domains.

Systematic Framework Design: The Ctrl&Shift architecture is designed with systematic and sound principles. The multi-task training (main task, removal, inpainting) is clearly motivated and helps the model disentangle the functions of different control signals (identity, location, pose), which is strongly validated by the ablation study (Table 3). The multi-stage training strategy (Stage 1 for priors, Stage 2 for real-world backgrounds) is logical. It first learns the core geometric knowledge on controllable synthetic data before generalizing to complex real-world scenes, effectively balancing geometric understanding and realism.

3.Strong Empirical Results and Evaluation:The method significantly outperforms existing SOTA approaches in both quantitative (Tables 1, 2) and qualitative (Fig. 4) comparisons.The evaluation metrics are comprehensive. They include not only traditional fidelity (PSNR) and identity preservation (DINO, CLIP) metrics but also creatively introduce geometric control accuracy metrics (Pose MAPE, Obj IoU), which are crucial for evaluating "controllable" generation tasks.The construction of the GeoEditBench benchmark is also a valuable contribution to the community, providing a standardized evaluation platform for this specific challenge.

**Weaknesses:**

I agree with the authors that this is excellent and inspiring work. To make the paper more complete and rigorous, I strongly recommend the authors add a 'Limitations and Future Work' discussion to the final version (e.g., in the conclusion or appendix).

I would like the authors to specifically address the following points:

From Technical Controllability to Practical Usability:
The authors should discuss the challenge of mapping intuitive user interactions (e.g., 2D mouse drags, rotational gestures) to the proposed 8D relative pose descriptor f. A powerful technology is of limited value without an intuitive interface.

Beyond Geometry: The Challenge of Physical Realism:
The authors should acknowledge that the current framework focuses primarily on geometric consistency, while the modeling of physical interactions (especially lighting, shadows, and reflections) is limited. Generating physically correct new shadows and reflections based on the new location's lighting conditions is a major challenge for this method (and all manipulation methods) and a valuable direction for future research.

Generalization Boundaries Induced by the Data Pipeline:
The authors should discuss potential failure cases arising from the dependency on the Image2Mesh pipeline. The method will likely struggle to generalize to: Non-rigid objects (e.g., cloth, hair, pets) Transparent or highly reflective objects (e.g., glass, metal)
Topologically complex objects (e.g., trees, smoke) Scenes with complex occlusion (e.g., moving an object behind another object in the scene; the current 'remove + inpaint' framework may not correctly handle this new depth relationship).

And I expect the open-source of the benchmark.

**Questions:**

As weakness.

---

> ### Author Response · Authors · 2025-11-21
> **Response to Reviewer 19Wi**
>
> # Response to Reviewer 19Wi
>
> We sincerely thank you for the thoughtful and comprehensive review, and for recognizing the conceptual innovation and systematic design of our work. We greatly appreciate your constructive suggestions for improving the completeness and rigor of the paper. Besides, we also show failure case of our framework in **Appendix A.8**.
>
> **Regarding Limitations and Future Work:** Following your recommendation, we have added a dedicated "Limitations and Future Work" section (Section 4) to the revised manuscript. We include this section here explicitly for easy reference:
>
> While our framework achieves precise geometry-aware object manipulation, we acknowledge several limitations stemming from both the data construction pipeline and the inherent complexity of the task (See Appendix A.8 for details). We discuss these challenges and future directions below:
>
> **From Technical Controllability to Practical Usability.** While our 8D relative pose descriptor $\mathbf{f}$ provides robust geometric control for the model, we recognize that manually specifying these values is not user-friendly. A key challenge for future work is to map intuitive user interactions, such as 2D mouse drags, rotational gestures, or manipulation via a 3D gizmo, directly to this descriptor. Since $\mathbf{f}$ represents a rigid transformation, it can be analytically derived from such interface inputs, bridging the gap between our precise internal representation and an accessible user experience.
>
> **Physical Realism Beyond Geometry.**
> Our current framework prioritizes geometric consistency. It does not explicitly model physical interactions such as lighting, variable shadows, or specular reflections. Instead, we rely on the model to implicitly learn these photometric effects from the training data. While often effective, this data-driven approach can struggle to generate physically correct shadows or reflections when the object is moved to a location with drastically different illumination conditions. Integrating explicit lighting estimation or physics-based rendering guidance remains a valuable direction to enhance realism.
>
> **Generalization Boundaries via the Data Pipeline.**
> Our method's reliance on 3D mesh reconstruction imposes specific generalization boundaries: **Object Type:** The pipeline assumes rigid geometry, limiting appli88cability to non-rigid objects (e.g., cloth, hair) or topological changes (e.g., smoke, fluids). **Material Properties**: Transparent or highly reflective objects (e.g., glass, mirrors) are often reconstructed with "baked-in" background textures, leading to artifacts during synthesis. **Complex Occlusions:** Our current "remove-and-inpaint" strategy assumes the target area is visible or planar. It cannot currently handle complex depth relationships, such as moving an object behind another scene element, as this would require reasoning about the geometry of the occluding background.
>
>
>
> **Regarding Benchmark Release:** We are fully committed to open-sourcing the GeoEditBench benchmark, including the evaluation protocol, data, and code. We believe this benchmark will provide valuable standardized evaluation for the community and facilitate future research in geometry-aware manipulation.
>
> We believe these additions significantly strengthen the paper by providing a transparent and comprehensive discussion of the method's scope, limitations, and promising directions for future research. Thank you again for the insightful suggestions that helped improve the completeness of our work.

---

> > ### Comment · Reviewer_19Wi · 2025-11-23
> >
> > I thank the authors for the rebuttal.
> >
> > I will keep my score.

---

> > > ### Author Response · Authors · 2025-11-24
> > > **Response to Reviewer 19Wi**
> > >
> > > Thank you for your thoughtful feedback and for maintaining your positive evaluation. We greatly appreciate your constructive suggestions regarding limitations and future work. Thanks again for all your effort and valuable input!

---

### Official Review · Reviewer_akfP · 2025-10-29

**Soundness:** 2
**Presentation:** 2
**Contribution:** 3
**Rating:** 6
**Confidence:** 3

**Summary:**

This paper addresses geometry-aware object manipulation in images and videos—specifically, relocating and reorienting objects while maintaining photorealism, background preservation, and geometric consistency under viewpoint changes. The proposed solution, Ctrl&Shift, is a diffusion framework that decomposes the task into object removal and reference-guided inpainting under explicit camera control. This approach avoids the need for explicit 3D representations (like meshes or NeRFs) at inference time.

To enable training on real-world data with geometric supervision, the authors developed a pipeline involving 3D mesh reconstruction, pose estimation via differentiable rendering, and a learned harmonization model (Object Pasting Model) to synthesize target views. The authors introduce a new benchmark, GeoEditBench, and outperform recent methods in both visual fidelity and geometric accuracy.

**Strengths:**

- Empirical Results and High-Quality Output. The quantitative results are compelling. On the new GeoEditBench (Table 2), Ctrl&Shift shows substantial improvements in geometric accuracy (17.70% Pose MAPE vs. 24.36% for the next best) while also achieving the highest fidelity scores (PSNR, DreamSim). The qualitative comparisons (Figure 4) are impressive, clearly demonstrating the model's superiority in handling complex rotations and perspective shifts where competitors fail.

- Significant Enabling Contributions (Data Pipeline and Benchmark). The pipeline for creating paired supervision from real-world data (Figure 3) is a major contribution, addressing the bottleneck of acquiring high-quality, pose-annotated data. Additionally, the introduction of GeoEditBench provides a valuable tool for the community to evaluate geometry-aware editing.

**Weaknesses:**

Several aspects require clarification or further investigation.

1. Reliance on 3D Supervision during Training and Generalization Limits. The paper emphasizes avoiding explicit 3D representations at inference. However, the training data generation (Section 2.5) heavily relies on explicit 3D reconstruction (Hunyuan3D-2) and differentiable rendering. The model's generalization is therefore constrained by the capabilities of the underlying 3D reconstruction method. Furthermore, the rigorous filtering (IoU ≥ 0.90 for pose estimation, L336) likely biases the dataset towards objects that are easy to reconstruct and align. The paper needs a more thorough analysis of the data pipeline's success rate, the diversity of the resulting dataset, and how the filtering affects the model's ability to handle complex geometries, materials (e.g., transparent or reflective), or intricate occlusions.

2. Simplistic Inference-Time Target Mask Estimation. During inference, the target mask $\hat{M}^{tgt}$ is estimated using simple heuristics: squaring the source mask, scaling by distance ratio, and shifting (Section 2.1). This approach is very coarse and does not account for changes in the object's silhouette due to rotation or perspective distortion. For significant viewpoint changes, this estimate could be highly inaccurate. The paper lacks a sensitivity analysis on how the accuracy of $\hat{M}^{tgt}$ affects the final output, and it is unclear how the model corrects for large inaccuracies in the estimated mask.

3. Evaluation of Harmonization, Lighting, and Shadows. Photorealistic manipulation requires accurate handling of lighting and shadows. The framework relies on a separately trained Object Pasting Model (A.5) to generate harmonized training data. However, there is no explicit evaluation of this component, nor of the main model's ability to generate plausible lighting and, crucially, realistic cast shadows in the edited scene. The current metrics (PSNR, DINO) do not measure physical correctness. The reliance on this specialized harmonization model during data synthesis might bottleneck the main model's understanding of complex lighting interactions.

4. Underdeveloped Video Manipulation Capabilities. The abstract and introduction prominently feature video manipulation. However, the quantitative evaluations are image-based. The qualitative video results (Appendix A.8) exhibit noticeable flickering and lower visual quality, which the authors acknowledge. The claims regarding high-quality geometry-aware video manipulation are not fully substantiated, and the limitations should be discussed more prominently in the main paper.

5. Lack of Failure Mode Analysis. Understanding when the model fails (e.g., specific object types, extreme pose changes, complex lighting) would provide a more complete picture of the method's limitations.

**Questions:**

1. Data Pipeline Statistics and Bias: What is the retention rate of the initial data corpus (e.g., Pexels videos) after the IoU ≥ 0.90 filtering step? Could you comment on whether this filtering introduces a bias towards simpler objects or viewpoints, and how the model performs on categories where 3D reconstruction typically struggles?

2. Sensitivity to Target Mask: How sensitive is the model's performance to the accuracy of the estimated target mask $\hat{M}^{tgt}$ during inference? Could you provide an oracle experiment showing the performance if the ground-truth target mask were available at inference?

3. Harmonization and Shadows: How does the framework ensure realistic shadow casting and lighting for the manipulated object? Are there examples where the object is moved between significantly different lighting environments (e.g., from shadow to direct light)?

4. Stage II Training Strategy: In Stage II, the main branch is frozen (L297). What is the motivation for this? Does this prevent the model from learning complex interactions (e.g., realistic shadow casting) that might require updates to the main diffusion pathway? Was fine-tuning the full model attempted?

5. Large Viewpoint Changes: How does the model handle large viewpoint changes where significant parts of the object, unseen in the source image, must be synthesized? Is the model relying purely on learned priors for this object-inpainting task?

---

> ### Author Response · Authors · 2025-11-21
> **Response to Reviewer akfP**
>
> # Response to Reviewer akfP
> Thank you for your detailed and constructive feedback. Your questions highlight important aspects of our method that deserve clearer explanation.
>
> ## 1. Regarding data pipeline statistics and potential bias:
>
> We appreciate your important concern regarding dataset filtering and potential bias. We acknowledge that our IoU ≥ 0.90 filtering criterion introduces selectivity in the training data. **However, this stringent filtering is essential for achieving the precise geometry editing that is the core objective of this work.** Without such quality control, the model would be trained on inconsistent geometric transformations, undermining its ability to learn accurate pose-conditioned manipulation. To provide transparency regarding the data processing pipeline, we will include detailed statistics in appendix **Appendix A.10** in the revised manuscript showing the retention rate at each filtering stage. This will help readers understand the scope and limitations of our training data.
>
> We acknowledge that this filtering strategy likely biases our dataset toward objects with well-defined geometry and reliable segmentation. Objects with challenging visual properties, such as transparent materials, complex reflective surfaces, or highly intricate geometries, are underrepresented, as these categories pose fundamental challenges for the upstream segmentation model (Grounded-SAM2). Additionally, our current pipeline has limited video manipulation capability, which we explicitly discuss as a limitation in our paper. We have added an explicit discussion of these limitations in the revised manuscript (Section 4) and included failure case analysis in Appendix A.8, noting that extending our approach to handle such challenging object categories remains an important direction for future work.
>
>
> ## 2. Regarding target mask estimation sensitivity:
>
> We appreciate your astute observation regarding the simplicity of our mask estimation heuristic. This coarse estimation strategy is deliberately chosen because **obtaining accurate target masks at inference time would require explicit 3D reconstruction, which is exactly what our framework aims to avoid.** The estimated target mask serves as **rough auxiliary information** indicating the approximate region where the object should appear, rather than a precise boundary constraint. The actual rendering region is learned by the model through large-scale paired training, guided by the relative pose descriptor $\mathbf{f}$. Through this training process, the model learns to generate appropriate object boundaries that respect the geometric transformation encoded in $\mathbf{f}$, even when the estimated target mask is imperfect.
>
> To demonstrate the robustness of our mask estimation strategy, we present an ablation study in Table 1. The results reveal several key insights: First, when training with estimated masks, **providing ground-truth masks at inference yields only marginal improvement (Row 2 vs. Row 1)**, as the model has learned to rely primarily on the pose descriptor $\mathbf{f}$ rather than precise mask boundaries for geometric manipulation. **That means our current deisgn is not sensitive to the target mask estimation.** Second, training and inference with ground-truth masks achieves the best performance (Row 3), as accurate masks provide richer geometric information that simplifies the learning task. However, this approach is impractical for real-world deployment where ground-truth masks are unavailable without explicit 3D reconstruction. Most importantly, Row 4 demonstrates that models trained on ground-truth masks suffer significant performance degradation when only estimated masks are available at inference, confirming that such models develop a strong dependency on precise mask information. In contrast, our approach (Row 1) maintains robust performance under realistic conditions, validating our design choice to train with estimated masks for practical applicability.
>
> **Table 1: Ablation Study on Target Mask Strategy**
>
> | Training Target Mask | Inference Target Mask | PSNR ↑ | DINO ↑ | CLIP ↑ | DreamSim ↓ | Pose MAPE ↓ | Obj IoU ↑ |
> |---------------|----------------|--------|--------|--------|------------|-------------|-----------|
> | Estimated | Estimated | 28.71 | 85.23 | 90.44 | 0.112 | 17.70% | 0.83 |
> | Estimated | GT | 28.95 | 85.51 | 90.48 | 0.110 | 17.35% | 0.84 |
> | GT | GT | **29.58** | **86.42** | **90.89** | **0.104** | **16.20%** | **0.86** |
> | GT | Estimated | 26.83 | 80.67 | 84.72 | 0.185 | 25.85% | 0.71 |

---

> > ### Author Response · Authors · 2025-11-21
> > **Response to Reviewer akfP**
> >
> > ## 3. Regarding harmonization, lighting, and shadow handling:
> >
> > We appreciate your observation regarding the absence of explicit lighting and shadow modeling in our framework. **Rather than employing dedicated constraints or modules for photometric effects, our approach learns to handle lighting and shadows implicitly through the data-driven training process on real image/video distributions.**
> >
> > The key to achieving photometric consistency lies in our training paradigm: **both the object pasting model and the final object manipulation model are trained with imperfect inputs that exhibit lighting and shadow inconsistencies (our synthetic data), while the training target is always real-world videos/images with authentic lighting and shadows**. Specifically, in the object pasting model, the input mesh renderings at the original camera pose typically have different lighting characteristics compared to the actual object appearance in the real video. Similarly, in the final manipulation model, the input video (output from the object pasting model with different camera pose) contains the object under lighting conditions that differ from those at the target pose.
> >
> > Crucially, because the training objective consistently requires matching real-world video appearance with authentic photometric properties, the models are forced to learn to **adapt the object's appearance from the reference to the target position with appropriate lighting and shadows**, rather than performing simple copy-and-paste operations. This implicit learning mechanism enables the models to generate plausible lighting effects, cast shadows, and photometric harmonization without requiring explicit physical modeling or hand-crafted constraints.
> >
> > ## 4. Regarding Stage II training strategy (freezing the main branch):
> >
> > We appreciate your question regarding our decision to freeze the main diffusion branch during Stage II training. This design choice is motivated by the significant difference in data characteristics between the two training stages:
> >
> > - **Stage I**: Large-scale, diverse dataset including both synthetic objects and real-world videos with uniform backgrounds. This stage teaches the model fundamental 3D object priors and pose-conditioned geometric manipulation via the pose descriptor $\mathbf{f}$.
> > - **Stage II**: Exclusively real-world video data with complex backgrounds and lighting. While richer in photometric detail, this dataset is substantially smaller and less diverse after quality filtering. This stage focuses on photometric harmonization and scene integration.
> >
> > Freezing the backbone during Stage II prevents the model from **overfitting to the narrower Stage II distribution** and catastrophically forgetting the broad geometric understanding acquired in Stage I. To validate this design choice, we conducted an ablation study comparing our approach (frozen main branch with trainable control branch) against fully fine-tuning both branches in Stage II, as shown in Table 2.
> >
> > The results demonstrate that while full fine-tuning achieves marginally better photometric quality (PSNR, DreamSim), it significantly degrades geometric accuracy (Pose MAPE: 17.70% vs. 21.45%) and object boundary preservation (Obj IoU: 0.83 vs. 0.79). This confirms our hypothesis that unrestricted fine-tuning on the smaller Stage II dataset causes the model to lose its robust geometric manipulation capabilities. Our frozen-backbone approach strikes a better balance, maintaining strong geometric control while still learning effective photometric harmonization through the trainable control branch.
> >
> > **Table 2: Ablation Study on Stage II Training Strategy (Zero-shot Evaluation on GeoEditBench)**
> >
> > | Stage II Strategy | PSNR ↑ | DINO ↑ | CLIP ↑ | DreamSim ↓ | Pose MAPE ↓ | Obj IoU ↑ |
> > |-------------------|--------|--------|--------|------------|-------------|-----------|
> > | Freeze main, train control branch | 28.71 | **85.23** | **90.44** | 0.112 | **17.70%** | **0.83** |
> > | Fine-tune main & control branch | **29.12** | 84.85 | 90.38 | **0.108** | 21.45% | 0.79 |

---

> > > ### Author Response · Authors · 2025-11-21
> > > **Response to Reviewer akfP**
> > >
> > > ## 5. Regarding large viewpoint changes and unseen regions:
> > >
> > > This is an excellent question that touches on a fundamental challenge in our task. When large viewpoint changes occur, significant portions of the object that were invisible in the source image must be synthesized—a capability that requires strong 3D geometric priors rather than simple 2D image manipulation.
> > >
> > > **This challenge is precisely why Stage I training is essential to our framework.** While pre-trained video generation models understand real-world video distributions well, they lack explicit knowledge about 3D object geometry and cross-view consistency. Stage I addresses this limitation through large-scale synthetic data training with comprehensive viewpoint coverage. **By uniformly sampling camera poses across the full range of possible transformations, including extreme angles where most of the object surface is unseen in the reference image, we bake the 3D cross-view knowledge into our model.** Specifically, our sampling strategy covers:
> > >
> > > - $\text{yaw} \in (-\pi, \pi]$: full 360° azimuthal coverage
> > > - $\text{pitch} \in [-\pi/2, \pi/2]$: complete elevation range
> > > - $\text{distance} \in (0.2, 10)$: practical range for common objects
> > >
> > > This ensures that extreme viewpoint changes, such as front-to-top transitions, are well-represented in the training distribution. The synthetic data contains perfect correspondence between different views of the same object, enabling the model to internalize **3D-consistent object representations and learn to generate unseen regions in a geometrically plausible manner**.
> > >
> > > After Stage I training, the model develops an internal understanding of how objects appear from different viewpoints and can leverage the pose descriptor $\mathbf{f}$ to guide the synthesis of previously unseen regions. The 8-dimensional descriptor $\mathbf{f} = [\text{aa}(\mathbf{R}\_{\text{rel}}); \mathbf{t}\_{\text{rel}}; \Delta r\_x; \Delta r\_y]^\top$, while compact, is mathematically sufficient to encode arbitrarily large camera movements, providing complete parameterization of the relative camera transformation.

---

### Official Review · Reviewer_kSVy · 2025-11-02

**Soundness:** 3
**Presentation:** 3
**Contribution:** 3
**Rating:** 8
**Confidence:** 3

**Summary:**

This paper introduces Ctrl&Shift, a novel end-to-end diffusion-based framework for geometry-aware object manipulation in images and videos. Unlike prior methods that either rely on explicit 3D reconstruction (e.g., NeRFs, 3DGS) or lack fine-grained geometric control (e.g., prompt- or mask-based diffusion editors), Ctrl&Shift achieves precise object relocation and rotation while preserving background integrity—without requiring explicit 3D modeling at inference time.
The core idea is to decompose object manipulation into two subtasks: (1) object removal, and (2) reference-guided inpainting under explicit camera pose control. These are unified within a single diffusion model trained via a multi-task, multi-stage strategy. The model conditions on: source frames, reference object image, source/target masks, and a relative camera pose descriptor (8D: axis-angle rotation, translation, and NDC screen shifts). This pose is encoded via Fourier features and injected through cross-attention.
To enhance generalization, the authors introduce a scalable pipeline for constructing real-world datasets with estimated relative camera poses. Extensive evaluations on fidelity, consistency, and controllability demonstrate superior performance over state-of-the-art methods in tasks like object relocation, rotation, removal, and inpainting.

**Strengths:**

1.	Innovative decomposition of object manipulation into removal and pose-controlled inpainting within a unified diffusion model, enabling fine-grained geometric control (e.g., precise relocation and rotation) without relying on explicit 3D representations like NeRF or Gaussians, which improves scalability and avoids per-scene optimization.
2.	The multi-task training approach effectively disentangles conditioning signals (background, object identity, camera pose), leading to interpretable and controllable edits that generalize to real-world content.
3.	The experimental evaluation is comprehensive, including quantitative results on two benchmarks, a detailed user study, and extensive ablation studies that validate the contribution of each proposed component (multi-stage training, auxiliary tasks). The creation of the high-quality GeoEditBench dataset is itself a significant contribution.

**Weaknesses:**

Heavy Reliance on Data Synthesis Pipeline: The method's performance is heavily dependent on the quality of the synthesized training data generated by the multi-step pipeline (mesh reconstruction → pose estimation → harmonization). Errors or artifacts introduced at any of these stages could propagate into the final model, potentially limiting its performance on objects or scenes that are challenging for these upstream components (e.g., transparent objects, complex textures, artistic images) and also for extreme viewpoint shifts or complex occlusions. The paper would benefit from a deeper analysis of failure cases stemming from the data pipeline.

**Questions:**

1.	Have you explored extending the framework to handle multiple objects simultaneously or interactive edits (e.g., user feedback loops during inference)?
2.	The relative pose descriptor f is an 8-dimensional vector. How does the model generalize to large, out-of-distribution relative camera movements (e.g., moving from a front view to a top-down view) that may be sparsely represented in the training data?

---

> ### Author Response · Authors · 2025-11-21
> **Response to Reviewer kSVy**
>
> # Response to Reviewer kSVy
>
> Thank you for your thoughtful comments and insightful questions. We appreciate your careful consideration of our work.
>
> ## 1.Regarding the data synthesis pipeline
>
> You raise an excellent point about error propagation through our pipeline. We fully acknowledge that failures in upstream components can affect the final model's performance. As you correctly identified, **transparent objects** present a particular challenge, our current mesh reconstruction treats transparency as opaque surfaces with background colors baked into the texture, and this limitation propagates to the trained model. Similarly, objects with **complex and intricate geometric** details may not be perfectly captured during the reconstruction stage. We have included a more detailed discussion of these limitations and their impact on model performance in the revised manuscript (Section 4). Additionally, we have included visual examples showcasing these failure cases in the appendix **Appendix A.8**. We believe this transparency about failure modes will be valuable for future work building upon our framework.
>
> ## 2.Regarding multi-object editing and interactive refinement
>
> We have not yet explored editing multi-object simultaneously in the current work, but we agree this is a very promising direction. Our current pipeline is able to perform sequential editing with multiple objects, and we include such visual results in the appendix **Appendix A.9**. Extending our framework to handle multiple objects simultaneously would require modifications including multiple object masks, multiple pose descriptors, and a mechanism for proper attribution and separation of edits. We believe this is technically feasible and represents an exciting avenue for future research.
> Regarding inference-time optimization through user feedback or inversion techniques: these approaches are complementary to our method. Our framework focuses on enabling the backbone model to perform geometrically-controlled modifications in a feed-forward manner, while inference-time optimization methods could be applied on top of our approach to further refine results. We view these as orthogonal contributions that could work synergistically.
>
>
> ## 3. Regarding generalization to large camera movements:
>
> We would like to clarify the mathematical completeness and coverage of our 8-dimensional pose descriptor. The descriptor $\mathbf{f} = [\text{aa}(\mathbf{R}\_{\text{rel}}); \mathbf{t}\_{\text{rel}}; \Delta r\_x; \Delta r\_y]^\top$ provides a complete parameterization of the relative camera transformation, including rotation, translation, and image-space shifts. To ensure comprehensive coverage of the pose space during training, our Stage I synthetic data generation process uniformly samples camera poses across the full range of possible transformations. Specifically:
>
> - $\text{yaw} \in (-\pi, \pi]$: full 360° azimuthal coverage
> - $\text{pitch} \in [-\pi/2, \pi/2]$: complete elevation range
> - $\text{distance} \in (0.2, 10)$: practical range for common objects (theoretically $(0, \infty)$, narrowed for practical real-world scenarios)
> - NDC shifts $(r_x, r_y) \in [-1, 1]^2$: full normalized device coordinate space
>
> This sampling strategy ensures that extreme viewpoint changes, such as front-to-top transitions, are well-represented in the training distribution. The 8-dimensional descriptor, while compact, is mathematically sufficient to encode arbitrarily large camera movements, including the challenging cases you mentioned.

---

> > ### Comment · Reviewer_kSVy · 2025-11-28
> >
> > Thank you for the responses which have addressed my concerns. Therefore I keep my initial score.

---

> > > ### Author Response · Authors · 2025-11-28
> > > **Response to Reviewer kSVy**
> > >
> > > Thank you for your thoughtful feedback and for maintaining your positive evaluation. We greatly appreciate your constructive suggestions regarding data construction pipeline and failure case analysis. Thanks again for all your effort and valuable input!

---

### Meta-Review · Area_Chair_WLDP · 2026-01-02

**Summary:**

Reviewers generally agree that the presented paradigm is an important contribution in the 3D editing space. Most concerns that were raised were addressed by the authors, including:
* conducting new experiments to compare to 3D editing approaches such as DiffusionHandles
* revising significant sections of the paper to tone down claims, move related work into the main paper, and clarifying the architectural setup
* adding a limitations and future work section which clearly point out issues of the presented work

The paper presents a nice approach to 3D object manipulation, and is therefore recommended for acceptance.

**Reviewer Concerns:**

All reviewer concerns appear to have been addressed during the rebuttal.

**Reviewer Scores:**

All reviewers indicated that they leaned towards accepting the paper (either before or after engaging with the authors).

---

### Decision · Program_Chairs · 2026-01-26

Accept (Poster)